

# Evolutionary analysis of chloroplast tRNA of Gymnosperm revealed the novel structural variation and evolutionary aspect

Ting-Ting Zhang[1,*], Yi-Kun Hou[1,*], Ting Yang[1], Shu-Ya Zhang[1], Ming Yue[1], Jianni Liu[2] and Zhonghu Li[1]

[1] Key Laboratory of Resource Biology and Biotechnology in Western China, Ministry of Education, College of Life Sciences, Northwest University, Xi'an, China
[2] Early Life Institute, State Key Laboratory of Continental Dynamics, Department of Geology, Northwest University, Xi'an, China
[*] These authors contributed equally to this work.

Corresponding authors
Jianni Liu, eliljn@nwu.edu.cn, liujianni@126.com
Zhonghu Li, lizhonghu@nwu.edu.cn

## ABSTRACT

Gymnosperms such as ginkgo, conifers, cycads, and gnetophytes are vital components of land ecosystems, and they have significant economic and ecologic value, as well as important roles as forest vegetation. In this study, we investigated the structural variation and evolution of chloroplast transfer RNAs (tRNAs) in gymnosperms. Chloroplasts are important organelles in photosynthetic plants. tRNAs are key participants in translation where they act as adapter molecules between the information level of nucleic acids and functional level of proteins. The basic structures of gymnosperm chloroplast tRNAs were found to have family-specific conserved sequences. The tRNA$\Psi$-loop was observed to contain a conforming sequence, i.e., U-U-C-N-A-N$_2$. In gymnosperms, tRNA$^{Ile}$ was found to encode a "CAU" anticodon, which is usually encoded by tRNA$^{Met}$. Phylogenetic analysis suggested that plastid tRNAs have a common polyphyletic evolutionary pattern, i.e., rooted in abundant common ancestors. Analyses of duplication and loss events in chloroplast tRNAs showed that gymnosperm tRNAs have experienced little more gene loss than gene duplication. Transition and transversion analysis showed that the tRNAs are iso-acceptor specific and they have experienced unequal evolutionary rates. These results provide new insights into the structural variation and evolution of gymnosperm chloroplast tRNAs, which may improve our comprehensive understanding of the biological characteristics of the tRNA family.

## INTRODUCTION

Gymnosperms originated in the Paleozoic Devonian Period (about 385 million years ago), and they are key groups in terms of the transformation from spore reproduction to seed reproduction in higher plants (*Gerrienne et al., 2004*; *Crisp & Cook, 2011*). According to the latest phylogenetic classification, gymnosperm species are divided into eight orders,

12 families, 84 genera, and more than 1,000 species (*Wang & Ran, 2014*). Gymnosperms include ginkgo, cycads, conifers, and gnetophytes, which are grown in forests as important timber species and they provide raw materials for human usage, such as fiber, resin, and tannin (*Christenhusz et al., 2010*). In addition, gymnosperms include some important threatened plants, where 40% are at high risk of extinction (*Forest et al., 2018*). Recent phylogenetic and evolutionary studies of gymnosperms have demonstrated the rapid evolution of mitochondrial (mt) genes and provided further evidence of sister relationship between conifers and Gnetales (*Ran, Gao & Wang, 2010*). The high levels of genetic diversity and population differentiation among the *Pinus* species in gymnosperms have been studied based on plastid DNA markers (*Liu et al., 2014*). Other studies have indicated patterns related to the physiological ecology, phylogenetic relationships, and population genetic structure of gymnosperm species (*Yu et al., 2014*; *Li et al., 2015*; *Dong et al., 2016*). However, these studies mainly considered the phylogeny and evolution at the whole populations level. Thus, the detailed evolutionary characteristics of gymnosperms still need to be elucidated.

Chloroplasts are the site of photosynthesis and of various essential metabolic pathways, e.g., fatty acid and amino acid biosynthesis and the assimilation of nitrogen, sulfur, and selenium (*Hoober, 2006*; *Des Marais, 2000 Knorr & Heimann, 2001*; *Pilon-Smits et al., 2002*; *Guo et al., 2007*; *Kretschmer, Croll & Kronstad, 2017*). It is generally recognized that chloroplasts are derived from proto-eukaryotic symbiotic cyanobacteria that internalized in eukaryotic cells (*Hiroki & Daisuke, 2018*) and evolved into central organelles. Chloroplasts have their own genome encoding about 100 proteins and they are maternally inherited organelles in most angiosperm plants (*Abdallah, Salamini & Leister, 2000*; *Heuertz et al., 2004*; *Civan et al., 2014*). Among gymnosperms, paternal plastid inheritance is the typical characteristic of conifers (*Fauré et al., 1994*; *Kaundun & Matsumoto, 2011*). Studies have shown that the chloroplast genome is quite conserved with an average evolutionary rate of 0.2–1. $0 \times 10^{-9}$ per site per year, which is only one-fifth of that for the nuclear genome (*Drouin, Daoud & Xia, 2008*; *Duchene & Bromham, 2013*). The chloroplast genome is a covalently closed circular structure with four parts comprising the large single copy region (LSC), small single copy region (SSC), inverted repeat region A (IRa), and inverted repeat region B (IRb). The two IRs have the same sequences but in the opposite direction (*Wang et al., 2008*; *Logacheva et al., 2009*; *Hereward et al., 2018*). Due to the independent evolution of the chloroplast genome, it is possible to construct a molecular phylogenetic tree using the chloroplast genome and without requiring any other data. Data analysis based on the conserved evolution of plastids is highly valuable for phylogenetic studies (*Kim & Suh, 2013*) because it can provide reliable and useful phylogenetic information. The relative completeness and independence of the chloroplast genome means that it can provide valuable material for research purposes.

Transfer RNAs (tRNAs) undergo numerous post-transcriptional nucleotide modifications and they exhibit abundant chemical diversity where the bases experience methylation, formylation, and other modifications (*Suzuki & Suzuki, 2014*). Chemical nucleotide modifications are frequent in tRNAs and they are important for the structure, stability, correct folding, aminoacylation, and decoding. For example, a previous analysis

of the chemically synthesized f⁵C34-modified anticodon loop of human mt-tRNA^Met showed that f⁵C34 contributes to the anticodon domain structure of the mt-tRNA (*Lusic et al., 2008*). tRNAs comprise sequences of less than 100 polynucleotides that fold into a clover-type secondary structure and then into an L-shaped tertiary structure (*Wilusz, 2015*). The secondary structure of tRNAs comprises different arms as well as loops, i.e., the D-arm, acceptor arm, anticodon arm, pseudouridine arm (Ψ-arm), D-loop, variable arm, anticodon loop, and pseudouridine loop (Ψ-loop) (*Giegé, Puglisi & Florentz, 1993*; *Mizutani & Goto, 2000*). This unique structure allows tRNA to act as important bridges between the information level of nucleic acids and functional level of proteins. The vital components of tRNAs comprise an anti-codon region that discerns the messenger RNA carried by the specific codons, a 3′-CCA tail for attaching to the cognate amino acid, the Ψ-arm, and a Ψ-loop that has a relationship with the ribosome machinery (*Kirchner & Ignatova, 2014*). Asymmetric combinations and the divided segments in tRNA genes allow us to understand the diversity of tRNA molecules. tRNA species fulfill various functions in cellular homeostasis, regulation of gene expression and epigenetics, biogenesis, and even biological disease (*Ribasd & Dedon, 2014*; *Kanai, 2015*; *Schimmel, 2017*). The evolutionary relationships determined between cyanobacteria and monocots show that tRNAs evolved polyphyletically and they originated from multiple common ancestors with a high rate of gene loss (*Mohanta et al., 2017*; *Mohanta et al., 2019*). Nevertheless, the basic details of the tRNAs in plant chloroplasts still need to be elucidated and the diverse evolutionary features of gymnosperm tRNAs are still unclear.

In this study, we assessed all of the chloroplast genomes in 12 families of gymnosperms from eight orders. The main aims of this study were as follows: (1) to determine the diversification of nucleotides in the secondary structure of gymnosperm tRNAs; (2) to identify the detailed genomic features of chloroplast tRNAs; (3) to assess the evolutionary relationships among different chloroplast tRNAs; and (4) to evaluate the duplication or loss events that occurred in all of the tRNAs considered. Our findings provide important insights into the biological characteristics and evolutionary variation of the tRNA family.

## MATERIALS & METHODS

### Annotation and identification of chloroplast tRNA sequences in gymnosperms

We downloaded complete chloroplast genomes for 12 representative gymnosperms in eight orders from the National Center for Biotechnology Information database (NCBI, https://www.ncbi.nlm.nih.gov/). The gymnosperm species investigated were: *Cycas debaoensis* Y. C. Zhong & C. J. Chen (KM459003), *Dioon spinulosum* Dyer ex Eichler (NC_027512), *Ginkgo biloba* L. (NC_016986), *Cedrus deodara* (Roxb.) G. Don (NC_014575), *Wollemia nobilis* W. G. Jones, K. D. Hill & J. M. Allen (NC_027235), *Retrophyllum piresii* Silba C. N. (KJ017081), *Sciadopitys verticillata* (Thunb.) Sieb. et Zucc. (NC_029734), *Cunninghamia lanceolata* (Lamb.) Hook. (NC_021437), *Taxus mairei* (Lemee et Levl.) Cheng et L. K. Fu (KJ123824), *Welwitschia mirabilis* Hook.f. (EU342371), *Gnetum gnemon* L. (KR476377), and *Ephedra equisetina* Bge. (NC_011954).

The gymnosperm tRNA genomes were annotated using GeSeq-Annotation of Organellar Genomes tool (*Tillich et al., 2017*) where the parameters were set as: circular sequence(s), chloroplast of sequence source, generate multi FASTA; BLAST protein search identity 25% for annotating plastid IR, 85% identity for BLAST rRNA, tRNA and DNA search, Embryophyta chloroplast (CDS+rRNA), third party tRNA annotator ARAGORN v1.2.38, ARWEN v1.2.3, tRNAScan-SE v2.0, and without Refseq choice.

## Structural analysis of chloroplast tRNAs

ARAGORN (*Laslett & Canback, 2004*) and tRNAScan-SE software (*Lowe & Eddy, 1997*) were employed to analyze the sequences and the secondary structure of tRNAs in the chloroplast genomes of the involved gymnosperm plants. The default parameters were set in ARAGORN software. The parameters for tRNAScan-SE were set as: sequence source, bacterial; search mode, default; query sequences, formatted (FASTA); and genetic code for tRNA isotype prediction, universal.

## Phylogenetic tree construction

A phylogenetic tree was constructed for all of the tRNAs using MEGA7.0 software (*Kumar et al., 2008*; *Kumar, Stecher & Tamura, 2016*). To study the evolutionary details of chloroplast tRNAs in gymnosperm species, an alignment file for tRNAs was achieved by CLUSTAL Omega software before the phylogenetic tree was constructed. MEGA7 software was used to transform the alignment file into MEGA format. The phylogenetic tree was constructed with the following parameters: phylogeny reconstruction of analysis, maximum likelihood model, bootstrap method in phylogeny test, 1,000 bootstrap replicates, nucleotides type, gamma distributed with invariant sites (G+I) model, five discrete gamma categories, partial deletion for gaps/missing data treatment, 95% site coverage cut-off, and very strong for branch swap filter.

## Transition/transversion analysis

The sequences of the tRNA isotypes were aligned to determine the transition and transversion rates for chloroplast tRNAs in gymnosperm plants. The files covering all 20 types of tRNAs were transformed into the MEGA file format and analyzed separately using MEGA7.0 software (*Kumar, Tamura & Nei, 1994*). The transition and transversion rates were analyzed for tRNAs with the following parameters: substitution pattern estimation (ML) analysis, automatic (neighbor-joining tree), maximum likelihood statistical method, nucleotide substitution type, Kimura two-parameter model, gamma distributed (G) site rates, five discrete gamma categories, partial deletion of gaps/missing data treatment, 95% of site coverage cut-off, and very strong branch swap filter.

## Loss and duplication events analysis for tRNA genes

In order to investigate the duplication or loss events in tRNA genes, the NCBI taxonomy browser was utilized to construct the whole species tree for the 12 gymnosperm species considered. The phylogenetic tree conducted in the evolutionary study was employed as gene tree. The gene tree for the tRNAs and species tree for the gymnosperm species were submitted to Notung 2.9 software (*Chen, Durand & Farach-Colton, 2000*), and then

**Table 1   A view of the gymnosperms in analysis.** Statistics of the 12 gymnosperms in the study.

| Order | Family | Subfamily | Genus | Species | NCBI Locus |
|---|---|---|---|---|---|
| Cycadales | Cycadaceae | | *Cycas* | *debaoensis* | KM459003 |
| | Zamiaceae | Diooideae | *Dioon* | *spinulosum* | NC_027512 |
| Ginkgoales | Ginkgoaceae | | *Ginkgo* | *biloba* | NC_016986 |
| Pinales | Pinaceae | Abieteae | *Cedrus* | *deodara* | NC_014575 |
| Araucariales | Araucariaceae | | *Wollemia* | *nobilis* | NC_027235 |
| | Podocarpaceae | | *Retrophyllum* | *piresii* | KJ017081 |
| Cupressales | Sciadopityaceae | | *Sciadopitys* | *verticillata* | NC_029734 |
| | Cupressaceae | Cunninghamia | *Cunninghamia* | *lanceolata* | NC_021437 |
| | Taxaceae | | *Taxus* | *mairei* | KJ123824 |
| Welwitschiales | Welwitschiaceae | | *Welwitschia* | *mirabilis* | EU342371 |
| Gnetales | Gnetaceae | | *Gnetum* | *gnemon* | KR476377 |
| Ephedrales | Ephedraceae | | *Ephedra* | *equisetina* | NC_011954 |

reconciled to discover duplicated and lost tRNA genes in the chloroplast genomes of gymnosperms.

## RESULTS

### Genomic features of gymnosperm chloroplast tRNAs

Sequences were analyzed to identify the genomic tRNAs in the chloroplast genomes of 12 gymnosperm species comprising *C. debaoensis*, *D. spinulosum*, *G. biloba*, *C. deodara*, *W. nobilis*, *R. piresii*, *S. verticillata*, *C. lanceolata*, *T. mairei*, *W. mirabilis*, *G. gnemon*, and *E. equisetina*, which were obtained from the NCBI database (Table 1). The results showed that the length of the chloroplast tRNAs vary from the smallest with 64 nucleotides (nt) (tRNA$^{Met}$ -CAU in *T. mairei*) to the largest with 96 nt (tRNA$^{Tyr}$-AUA in *W. nobilis*, *C. deodara*, and *G. biloba*) (Data S1). We found that the chloroplast genomes of gymnosperm plants encode 28 to 33 tRNAs (Table 2), where *D. spinulosum*, *C. deodara*, and *S. verticillata* encode 31 anticodons, *W. nobilis*, *R. piresii*, *C. lanceolata*, and *G. gnemon* encode 32 tRNA isotypes, *G. biloba*, and *W. mirabilis* encode 33 tRNAs. Other species comprising *T. mairei*, *E. equisetina* and *C. debaoensis* encode 28, 28, 30 tRNA isotypes, respectively (Table 2). tRNA$^{Ala}$ was not found in *R. piresii* and *T. mairei*, and tRNA$^{Val}$ was not detected in *T. mairei* (Fig. S3). We also observed that all of the species do not encode selenocysteine and its suppressor tRNA (Table 2). Overall, tRNA$^{Ser}$ (in *W. nobilis*) and tRNA$^{Arg}$ (in *W. mirabilis*) are the most abundant (four types) followed by tRNA$^{Leu}$ (three types) (Table 2).

### Variations in structures of chloroplast tRNAs

Some tRNAs with a loop structure in the variable region were found to be encoded in the gymnosperm chloroplast genomes (Figs. 1 and 2). A novel tRNA lacking the D-arm was found in tRNA$^{Gly}$ in *W. nobilis* (Fig. 3). As shown in Figs. 1 and 2, tRNA$^{Leu}$, tRNA$^{Ser}$, and tRNA$^{Tyr}$ contain expanded variable stem/loops. In these tRNAs (except for tRNA$^{Ser}$-GCU of *D. spinulosum*), the anticodon loop of tRNA$^{Ser}$ contains the conserved consensus

Zhang et al. (2020), *PeerJ*, DOI 10.7717/peerj.10312

**Table 2** Distribution of tRNA isotypes in chloroplast genome of gymnosperms.

| tRNA isotypes | Number of tRNAs | | | | | | | | | | | |
|---|---|---|---|---|---|---|---|---|---|---|---|---|
| | *C. debaoensis* | *D. spinulosum* | *G. biloba* | *C. deodara* | *W. nobilis* | *R. piresii* | *S. verticillata* | *C. lanceolata* | *T. mairei* | *W. mirabilis* | *G. gnemon* | *E. equisetina* |
| Ala | 1 | 1 | 1 | 1 | 1 | 0 | 1 | 1 | 0 | 1 | 1 | 1 |
| Gly | 1 | 1 | 2 | 2 | 2 | 2 | 1 | 1 | 1 | 2 | 2 | 1 |
| Pro | 2 | 2 | 2 | 2 | 2 | 2 | 1 | 2 | 2 | 2 | 2 | 1 |
| Thr | 1 | 2 | 2 | 2 | 2 | 2 | 2 | 2 | 2 | 2 | 2 | 1 |
| Val | 2 | 2 | 1 | 1 | 1 | 2 | 2 | 2 | 0 | 2 | 2 | 1 |
| Ser | 3 | 3 | 3 | 3 | 4 | 3 | 3 | 3 | 2 | 3 | 3 | 3 |
| Arg | 3 | 3 | 3 | 3 | 3 | 3 | 3 | 2 | 2 | 4 | 3 | 2 |
| Leu | 3 | 3 | 3 | 3 | 3 | 3 | 3 | 3 | 3 | 3 | 3 | 3 |
| Phe | 1 | 1 | 1 | 1 | 1 | 1 | 1 | 1 | 1 | 1 | 1 | 1 |
| Asn | 1 | 1 | 1 | 1 | 1 | 1 | 1 | 1 | 1 | 1 | 1 | 1 |
| Lys | 1 | 1 | 1 | 1 | 1 | 1 | 1 | 2 | 1 | 1 | 1 | 1 |
| Asp | 1 | 1 | 1 | 1 | 1 | 2 | 1 | 1 | 1 | 1 | 1 | 1 |
| Glu | 2 | 2 | 1 | 2 | 2 | 2 | 2 | 2 | 1 | 2 | 2 | 2 |
| His | 1 | 1 | 2 | 1 | 1 | 1 | 1 | 1 | 1 | 1 | 1 | 1 |
| Gln | 1 | 1 | 1 | 1 | 1 | 1 | 2 | 2 | 1 | 1 | 1 | 1 |
| Ile | 1 | 1 | 1 | 1 | 1 | 1 | 1 | 1 | 4 | 1 | 1 | 1 |
| Met/fMet | 2 | 2 | 2 | 2 | 2 | 2 | 2 | 2 | 2 | 2 | 2 | 2 |
| Tyr | 1 | 1 | 2 | 1 | 1 | 1 | 1 | 1 | 1 | 1 | 1 | 1 |
| Cys | 1 | 1 | 2 | 1 | 1 | 1 | 1 | 1 | 1 | 1 | 1 | 1 |
| Trp | 1 | 1 | 1 | 1 | 1 | 1 | 1 | 1 | 1 | 1 | 1 | 2 |
| Selenocysteine | 0 | 0 | 0 | 0 | 0 | 0 | 0 | 0 | 0 | 0 | 0 | 0 |
| Suppressor | 0 | 0 | 0 | 0 | 0 | 0 | 0 | 0 | 0 | 0 | 0 | 0 |
| Total | 30 | 31 | 33 | 31 | 32 | 32 | 31 | 32 | 28 | 33 | 32 | 28 |

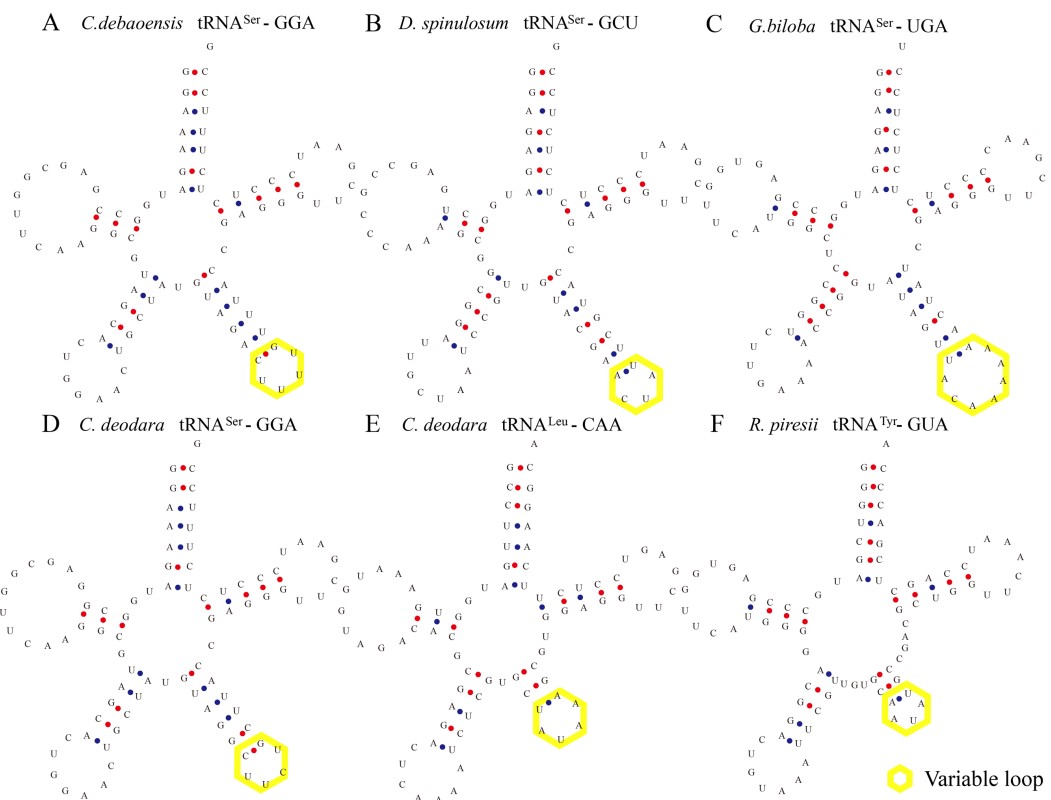

**Figure 1** **Certain tRNAs in *C. debaoensis*, *D. spinulosum*, *G. biloba*, *C. deodara*, and *R. piresii* contain expanded variable stem and loops.** tRNASer, tRNALeu, and tRNATyr from *C. debaoensis* (A, tRNASer-GGA), *D. spinulosum* (B, tRNASer-GCU), *G. biloba* (C, tRNASer-UGA), *C. deodara* (D, tRNASer-GGA; E, tRNALeu-CAA), *R. piresii* (F, tRNATyr-GUA) were observed to contain an expanded variable stem and variable loop (indicated by yellow box). The anti-codon loop of tRNASer (except for tRNASer-GCU of *D. spinulosum*) was made up of seven nucleotides with the conservative N-U-N-G-A-A-N consensus sequence.

sequence N-U-N-G-A-A-N, and tRNAs$^{Leu}$ have the consensus sequence C-U-N-A-N$_2$-A. The variable loop region is predicted to fold into stem-loop structures with apical loops of 3 to 7 nt in tRNA$^{Ser}$ and several tRNA$^{Leu}$ variants. The stems contain up to 7 bp (Figs. 1 and 2). The expanded variable loop structures may play important functions during the protein translation process in chloroplasts.

## Chloroplast genomes contain 25 to 30 anticodon-specific tRNAs

The genomes of the species analyzed were found to code for at least two copies of tRNA$^{Met}$-CAU/tRNA$^{fMet}$-CAU. Each of the gymnosperm chloroplast genomes encodes 25 to 30 anticodon-specific tRNAs (Tables 2 and 3), where *E. equisetina* encodes 25 anticodons, *T. mairei* encodes 26 anticodons, *C. debaoensis*, *S. verticillata*, and *C. lanceolata* encode 28 anticodons, and *D. spinulosum*, *C. deodara*, *W. mirabilis*, and *R. piresii* encode 29 anticodons. Other species comprising *W. nobilis*, *G. gnemon* and *G. biloba* encodes 30 anticodons (Table 3).

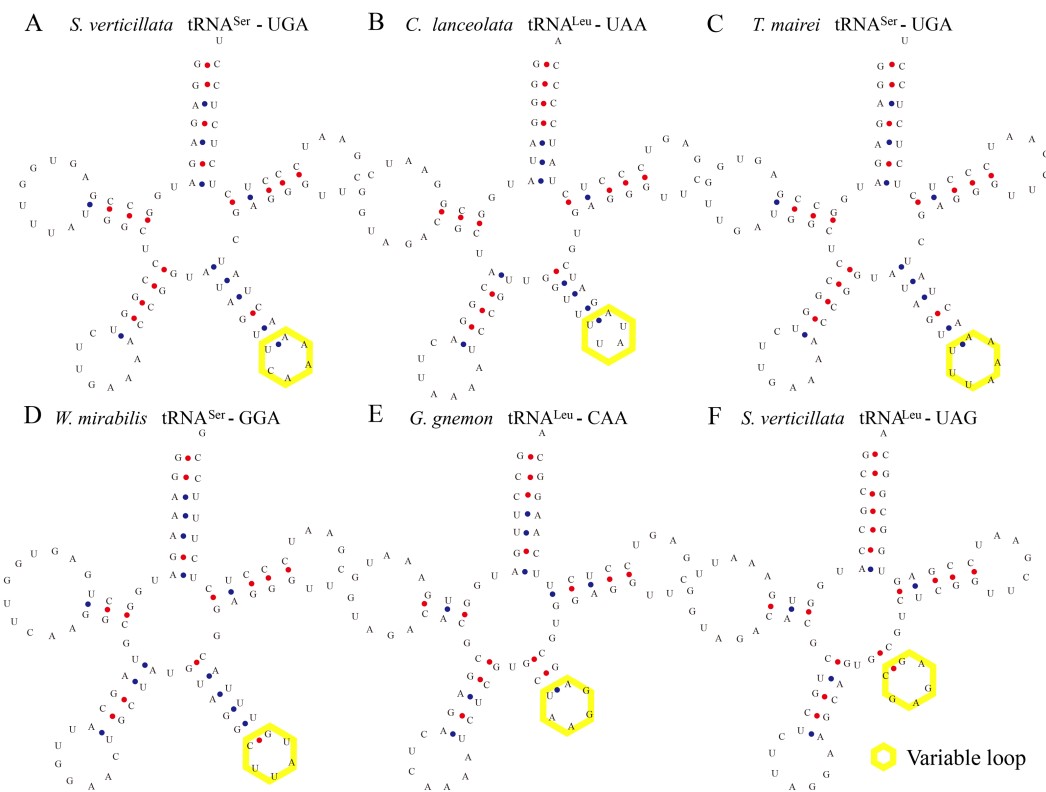

**Figure 2** **Certain tRNAs in *S. verticillata, C. lanceolata, T. mairei, W. mirabilis,* and *G. gnemon* contain expanded variable stem and loops.** tRNA$_{Ser}$, tRNA$_{Leu}$ from *S. verticillate* (A, tRNA$_{Ser}$-UGA; F, tRNA$_{Leu}$-UAG), *C. lanceolata* (B, tRNA$_{Leu}$-UAA), *T. mairei* (C, tRNA$_{Ser}$-UGA), *W. mirabilis* (D, tRNA$_{Ser}$-GGA), *G. gnemon* (E, tRNA$_{Leu}$-CAA) were observed to contain a variable stem and variable loop (indicated by yellow box). The anti-codon loop of tRNA$_{Ser}$ was made up of seven nucleotides with the conservative N-U-N-G-A-A-N consensus sequence, and the consensus sequence was C-U-N-A-N2-A for tRNA$_{Leu}$.

tRNA$^{Arg}$-CCG was present in the genomes of nine gymnosperm species but absent from *C. lanceolata*, *T. mairei*, and *E. equisetina*, while tRNA$^{Gly}$-UCC was lacking from *C. debaoensis*, *S. verticillata*, *D. spinulosum*, *C. lanceolata*, *T. mairei*, and *E. equisetina* (Table 3). The most abundant anticodons found in the chloroplast genomes were tRNA$^{Gly}$-GCC, tRNA$^{Pro}$-UGG, tRNA$^{Ser}$-UGA, tRNA$^{Ser}$-GCU, tRNA$^{Arg}$-ACG, tRNA$^{Arg}$-UCU, tRNA$^{Leu}$-UAG, tRNA$^{Leu}$-CAA, tRNA$^{Phe}$-GAA, tRNA$^{Asn}$-GUU, tRNA$^{Lys}$-UUU, tRNA$^{Asp}$-GUC, tRNA$^{Glu}$-UUC, tRNA$^{His}$-GUG, tRNA$^{Gln}$-UUG, tRNA$^{Ile}$-CAU, tRNA$^{Met}$-CAU, tRNA$^{Tyr}$-GUA, tRNA$^{Cys}$-GCA, and tRNA$^{Trp}$-CCA (Table 3). Two tRNA$^{Trp}$ iso-acceptors are present in *E. equisetina* chloroplasts, compared with a single one in the other gymnosperm species analyzed in this study.

## Conserved gymnosperm chloroplast tRNAs

The clover leaf-like secondary structure of a tRNA is shown in Fig. 4. In the study, we found that most tRNAs contain a "G" as the first nucleotide in the D-arm, except for tRNA$^{Lys}$, tRNA$^{Met}$, tRNA$^{Pro}$, tRNA$^{Thr}$, tRNA$^{Tyr}$, and tRNA$^{Val}$. "A" is present in the first and the last position of the D-loop apart from tRNA$^{Gly}$, tRNA$^{Ile}$, tRNA$^{Leu}$, tRNA$^{Met}$, and

Zhang et al. (2020), *PeerJ*, DOI 10.7717/peerj.10312

**Table 3  Distribution of anti-codons in the chloroplast genome of gymnosperms.** Each of the gymnosperm chloroplast genomes encodes 25 to 30 anticodon-specific tRNAs. *E. equisetina* encodes 25 anticodons, *T. mairei* encodes 26 anticodons, *C. debaoensis*, *S. verticillata*, and *C. lanceolata* encode 28 anticodons, and *D. spinulosum*, *C. deodara*, *W. mirabilis*, and *R. piresii* encode 29 anticodons. Other species comprising *W. nobilis*, *G. gnemon* and *G. biloba* encodes 30 anticodons.

| tRNA Isotypes | Isoacceptors | | | | | | tRNA Isotypes | Isoacceptors | | | | | |
|---|---|---|---|---|---|---|---|---|---|---|---|---|---|
| | *C. debaoensis* (28) | | | | | | | *S. verticillata* (28) | | | | | |
| Ala | AGC: 0 | GGC: 0 | CGC: 0 | UGC: 1 | | | Ala | AGC: 0 | GGC: 0 | CGC: 0 | UGC: 1 | | |
| Gly | ACC: 0 | GCC: 1 | CCC: 0 | UCC: 0 | | | Gly | ACC: 0 | GCC: 1 | CCC: 0 | UCC: 0 | | |
| Pro | AGG: 0 | GGG: 1 | CGG: 0 | UGG: 1 | | | Pro | AGG: 0 | GGG: 0 | CGG: 0 | UGG: 1 | | |
| Thr | AGU: 0 | GGU: 0 | CGU: 0 | UGU: 1 | | | Thr | AGU: 0 | GGU: 1 | CGU: 0 | UGU: 1 | | |
| Val | AAC: 0 | GAC: 1 | CAC: 0 | UAC: 1 | | | Val | AAC: 0 | GAC: 1 | CAC: 0 | UAC: 1 | | |
| Ser | AGA: 0 | GGA: 1 | CGA: 0 | UGA: 1 | ACU: 0 | GCU: 1 | Ser | AGA: 0 | GGA: 1 | CGA: 0 | UGA: 1 | ACU: 0 | GCU: 1 |
| Arg | ACG: 1 | GCG: 0 | CCG:1 | UCG: 0 | CCU: 0 | UCU: 1 | Arg | ACG: 1 | GCG: 0 | CCG:1 | UCG: 0 | CCU: 0 | UCU: 1 |
| Leu | AAG: 0 | GAG: 0 | CAG: 0 | UAG: 1 | CAA: 1 | UAA: 1 | Leu | AAG: 0 | GAG: 0 | CAG: 0 | UAG: 1 | CAA: 1 | UAA: 1 |
| Phe | AAA: 0 | GAA: 1 | | | | | Phe | AAA: 0 | GAA: 1 | | | | |
| Asn | AUU: 0 | GUU: 1 | | | | | Asn | AUU: 0 | GUU: 1 | | | | |
| Lys | CUU: 0 | UUU: 1 | | | | | Lys | CUU: 0 | UUU: 1 | | | | |
| Asp | AUC: 0 | GUC: 1 | | | | | Asp | AUC: 0 | GUC: 1 | | | | |
| Glu | CUC: 0 | UUC: 2 | | | | | Glu | CUC: 0 | UUC: 2 | | | | |
| His | AUG: 0 | GUG: 1 | | | | | His | AUG: 0 | GUG: 1 | | | | |
| Gln | CUG: 0 | UUG: 1 | | | | | Gln | CUG: 0 | UUG: 2 | | | | |
| Ile | AAU: 0 | GAU: 0 | CAU: 1 | UAU: 0 | | | Ile | AAU: 0 | GAU: 0 | CAU: 1 | UAU: 0 | | |
| Met | CAU: 2 | | | | | | Met | CAU: 2 | | | | | |
| Tyr | AUA: 0 | GUA: 1 | | | | | Tyr | AUA: 0 | GUA: 1 | | | | |
| Cys | ACA: 0 | GCA: 1 | | | | | Cys | ACA: 0 | GCA: 1 | | | | |
| Trp | CCA: 1 | | | | | | Trp | CCA: 1 | | | | | |
| Supressor | CUA: 0 | UUA: 0 | UCA: 0 | | | | Supressor | CUA: 0 | UUA: 0 | UCA: 0 | | | |
| Sec | UCA: 0 | | | | | | Sec | UCA: 0 | | | | | |
| | *D. spinulosum* (29) | | | | | | | *C. lanceolata* (28) | | | | | |
| Ala | AGC: 0 | GGC: 0 | CGC: 0 | UGC: 1 | | | Ala | AGC: 0 | GGC: 0 | CGC: 0 | UGC: 1 | | |
| Gly | ACC: 0 | GCC: 1 | CCC: 0 | UCC: 0 | | | Gly | ACC: 0 | GCC: 1 | CCC: 0 | UCC: 0 | | |
| Pro | AGG: 0 | GGG: 1 | CGG: 0 | UGG: 1 | | | Pro | AGG: 0 | GGG: 1 | CGG: 0 | UGG: 1 | | |
| Thr | AGU: 0 | GGU: 1 | CGU: 0 | UGU: 1 | | | Thr | AGU: 0 | GGU: 1 | CGU: 0 | UGU: 1 | | |
| Val | AAC: 0 | GAC: 1 | CAC: 0 | UAC: 1 | | | Val | AAC: 0 | GAC: 1 | CAC: 0 | UAC: 1 | | |
| Ser | AGA: 0 | GGA: 1 | CGA: 0 | UGA: 1 | ACU: 0 | GCU: 1 | Ser | AGA: 0 | GGA: 1 | CGA: 0 | UGA: 1 | ACU: 0 | GCU: 1 |

| tRNA Isotypes | Isoacceptors | | | | | |
|---|---|---|---|---|---|---|
| Arg | ACG: 1 | GCG: 0 | CCG:1 | UCG: 0 | CCU: 0 | UCU: 1 |
| Leu | AAG: 0 | GAG: 0 | CAG: 0 | UAG: 1 | CAA: 1 | UAA: 1 |
| Phe | AAA: 0 | GAA: 1 | | | | |
| Asn | AUU: 0 | GUU: 1 | | | | |
| Lys | CUU: 0 | UUU: 1 | | | | |
| Asp | AUC: 0 | GUC: 1 | | | | |
| Glu | CUC: 0 | UUC: 2 | | | | |
| His | AUG: 0 | GUG: 2 | | | | |
| Gln | CUG: 0 | UUG: 1 | | | | |
| Ile | AAU: 0 | GAU: 0 | CAU: 1 | UAU: 0 | | |
| Met | CAU: 2 | | | | | |
| Tyr | AUA: 0 | GUA: 1 | | | | |
| Cys | ACA: 0 | GCA: 1 | | | | |
| Trp | CCA: 1 | | | | | |
| Supressor | CUA: 0 | UUA: 0 | UCA: 0 | | | |
| Sec | UCA: 0 | | | | | |

| tRNA Isotypes | Isoacceptors | | | | | |
|---|---|---|---|---|---|---|
| Arg | ACG: 1 | GCG: 0 | CCG: 0 | UCG: 0 | CCU: 0 | UCU: 1 |
| Leu | AAG: 0 | GAG: 0 | CAG: 0 | UAG: 1 | CAA: 1 | UAA: 1 |
| Phe | AAA: 0 | GAA: 1 | | | | |
| Asn | AUU: 0 | GUU: 1 | | | | |
| Lys | CUU: 1 | UUU: 1 | | | | |
| Asp | AUC: 0 | GUC: 1 | | | | |
| Glu | CUC: 0 | UUC: 2 | | | | |
| His | AUG: 0 | GUG: 1 | | | | |
| Gln | CUG: 0 | UUG: 2 | | | | |
| Ile | AAU: 0 | GAU: 0 | CAU: 1 | UAU: 0 | | |
| Met | CAU: 2 | | | | | |
| Tyr | AUA: 0 | GUA: 1 | | | | |
| Cys | ACA: 0 | GCA: 1 | | | | |
| Trp | CCA: 1 | | | | | |
| Supressor | CUA: 0 | UUA: 0 | UCA: 0 | | | |
| Sec | UCA: 0 | | | | | |

*G. biloba* (30)

| tRNA Isotypes | Isoacceptors | | | | | |
|---|---|---|---|---|---|---|
| Ala | AGC: 0 | GGC: 0 | CGC: 0 | UGC: 1 | | |
| Gly | ACC: 0 | GCC: 1 | CCC: 0 | UCC: 1 | | |
| Pro | AGG: 0 | GGG: 1 | CGG: 0 | UGG: 1 | | |
| Thr | AGU: 0 | GGU: 1 | CGU: 0 | UGU: 1 | | |
| Val | AAC: 0 | GAC: 1 | CAC: 0 | UAC: 0 | | |
| Ser | AGA: 0 | GGA: 1 | CGA: 0 | UGA: 1 | ACU: 0 | GCU: 1 |
| Arg | ACG: 1 | GCG: 0 | CCG:1 | UCG: 0 | CCU: 0 | UCU: 1 |
| Leu | AAG: 0 | GAG: 0 | CAG: 0 | UAG: 1 | CAA: 2 | UAA: 0 |
| Phe | AAA: 0 | GAA: 1 | | | | |
| Asn | AUU: 0 | GUU: 1 | | | | |
| Lys | CUU: 0 | UUU: 1 | | | | |
| Asp | AUC: 0 | GUC: 1 | | | | |
| Glu | CUC: 0 | UUC: 1 | | | | |
| His | AUG: 0 | GUG: 2 | | | | |
| Gln | CUG: 0 | UUG: 1 | | | | |
| Ile | AAU: 0 | GAU: 0 | CAU: 1 | UAU: 0 | | |
| Met | CAU: 2 | | | | | |
| Tyr | AUA: 1 | GUA: 1 | | | | |
| Cys | ACA: 1 | GCA: 1 | | | | |
| Trp | CCA: 1 | | | | | |

*T. mairei* (26)

| tRNA Isotypes | Isoacceptors | | | | | |
|---|---|---|---|---|---|---|
| Ala | AGC: 0 | GGC: 0 | CGC: 0 | UGC: 0 | | |
| Gly | ACC: 0 | GCC: 1 | CCC: 0 | UCC: 0 | | |
| Pro | AGG: 0 | GGG: 1 | CGG: 0 | UGG: 1 | | |
| Thr | AGU: 0 | GGU: 1 | CGU: 0 | UGU: 1 | | |
| Val | AAC: 0 | GAC: 0 | CAC: 0 | UAC: 0 | | |
| Ser | AGA: 0 | GGA: 0 | CGA: 0 | UGA: 1 | ACU: 0 | GCU: 1 |
| Arg | ACG: 1 | GCG: 0 | CCG: 0 | UCG: 0 | CCU: 0 | UCU: 1 |
| Leu | AAG: 0 | GAG: 0 | CAG: 0 | UAG: 1 | CAA: 1 | UAA: 1 |
| Phe | AAA: 0 | GAA: 1 | | | | |
| Asn | AUU: 0 | GUU: 1 | | | | |
| Lys | CUU: 0 | UUU: 1 | | | | |
| Asp | AUC: 0 | GUC: 1 | | | | |
| Glu | CUC: 0 | UUC: 1 | | | | |
| His | AUG: 0 | GUG: 1 | | | | |
| Gln | CUG: 0 | UUG: 1 | | | | |
| Ile | AAU: 1 | GAU: 0 | CAU: 2 | UAU: 1 | | |
| Met | CAU: 2 | | | | | |
| Tyr | AUA: 0 | GUA: 1 | | | | |
| Cys | ACA: 0 | GCA: 1 | | | | |
| Trp | CCA: 1 | | | | | |

Zhang et al. (2020), *PeerJ*, DOI 10.7717/peerj.10312

Zhang et al. (2020), *PeerJ*, DOI 10.7717/peerj.10312

**Table 3** (*continued*)

| tRNA Isotypes | Isoacceptors | | | | | | tRNA Isotypes | Isoacceptors | | | | | |
|---|---|---|---|---|---|---|---|---|---|---|---|---|---|
| Supressor | CUA: 0 | UUA: 0 | UCA: 0 | | | | Supressor | CUA: 0 | UUA: 0 | UCA: 0 | | | |
| Sec | UCA: 0 | | | | | | Sec | UCA: 0 | | | | | |
| | *C. deodara* (29) | | | | | | | *W. mirabilis* (29) | | | | | |
| Ala | AGC: 0 | GGC: 0 | CGC: 0 | UGC: 1 | | | Ala | AGC: 0 | GGC: 0 | CGC: 0 | UGC: 1 | | |
| Gly | ACC: 0 | GCC: 1 | CCC: 0 | UCC: 1 | | | Gly | ACC: 0 | GCC: 1 | CCC: 0 | UCC: 1 | | |
| Pro | AGG: 0 | GGG: 1 | CGG: 0 | UGG: 1 | | | Pro | AGG: 0 | GGG: 1 | CGG: 0 | UGG: 1 | | |
| Thr | AGU: 0 | GGU: 1 | CGU: 0 | UGU: 1 | | | Thr | AGU: 0 | GGU: 1 | CGU: 0 | UGU: 1 | | |
| Val | AAC: 0 | GAC: 1 | CAC: 0 | UAC: 0 | | | Val | AAC: 0 | GAC: 1 | CAC: 0 | UAC: 1 | | |
| Ser | AGA: 0 | GGA: 1 | CGA: 0 | UGA: 1 | ACU: 0 | GCU: 1 | Ser | AGA: 0 | GGA: 1 | CGA: 0 | UGA: 1 | ACU: 0 | GCU: 1 |
| Arg | ACG: 1 | GCG: 0 | CCG:1 | UCG: 0 | CCU: 0 | UCU: 1 | Arg | ACG: 1 | GCG: 0 | CCG: 2 | UCG: 0 | CCU: 0 | UCU: 1 |
| Leu | AAG: 0 | GAG: 0 | CAG: 0 | UAG: 1 | CAA: 1 | UAA: 1 | Leu | AAG: 0 | GAG: 0 | CAG: 0 | UAG: 1 | CAA: 1 | UAA: 1 |
| Phe | AAA: 0 | GAA: 1 | | | | | Phe | AAA: 0 | GAA: 1 | | | | |
| Asn | AUU: 0 | GUU: 1 | | | | | Asn | AUU: 0 | GUU: 1 | | | | |
| Lys | CUU: 0 | UUU: 1 | | | | | Lys | CUU: 0 | UUU: 1 | | | | |
| Asp | AUC: 0 | GUC: 1 | | | | | Asp | AUC: 0 | GUC: 1 | | | | |
| Glu | CUC: 0 | UUC: 2 | | | | | Glu | CUC: 0 | UUC: 2 | | | | |
| His | AUG: 0 | GUG: 1 | | | | | His | AUG: 0 | GUG: 1 | | | | |
| Gln | CUG: 0 | UUG: 1 | | | | | Gln | CUG: 0 | UUG: 1 | | | | |
| Ile | AAU: 0 | GAU: 04 | CAU: 1 | UAU: 0 | | | Ile | AAU: 0 | GAU: 0 | CAU: 1 | UAU: 0 | | |
| Met | CAU: 2 | | | | | | Met | CAU: 2 | | | | | |
| Tyr | AUA: 0 | GUA: 1 | | | | | Tyr | AUA: 0 | GUA: 1 | | | | |
| Cys | ACA: 0 | GCA: 1 | | | | | Cys | ACA: 0 | GCA: 1 | | | | |
| Trp | CCA: 1 | | | | | | Trp | CCA: 1 | | | | | |
| Supressor | CUA: 0 | UUA: 0 | UCA: 0 | | | | Supressor | CUA: 0 | UUA: 0 | UCA: 0 | | | |
| Sec | UCA: 0 | | | | | | Sec | UCA: 0 | | | | | |
| | *W. nobilis* (30) | | | | | | | *G. gnemon* (30) | | | | | |
| Ala | AGC: 0 | GGC: 0 | CGC: 0 | UGC: 1 | | | Ala | AGC: 0 | GGC: 0 | CGC: 0 | UGC: 1 | | |
| Gly | ACC: 0 | GCC: 1 | CCC: 0 | UCC: 1 | | | Gly | ACC: 0 | GCC: 1 | CCC: 0 | UCC: 1 | | |
| Pro | AGG: 0 | GGG: 1 | CGG: 0 | UGG: 1 | | | Pro | AGG: 0 | GGG: 1 | CGG: 0 | UGG: 1 | | |
| Thr | AGU: 0 | GGU: 1 | CGU: 0 | UGU: 1 | | | Thr | AGU: 0 | GGU: 1 | CGU: 0 | UGU: 1 | | |
| Val | AAC: 0 | GAC: 1 | CAC: 0 | UAC: 0 | | | Val | AAC: 0 | GAC: 1 | CAC: 0 | UAC: 1 | | |
| Ser | AGA: 0 | GGA: 1 | CGA: 1 | UGA: 1 | ACU: 0 | GCU: 1 | Ser | AGA: 0 | GGA: 1 | CGA: 0 | UGA: 1 | ACU: 0 | GCU: 1 |
| Arg | ACG: 1 | GCG: 0 | CCG:1 | UCG: 0 | CCU: 0 | UCU: 1 | Arg | ACG: 1 | GCG: 0 | CCG:1 | UCG: 0 | CCU: 0 | UCU: 1 |
| Leu | AAG: 0 | GAG: 0 | CAG: 0 | UAG: 1 | CAA: 1 | UAA: 1 | Leu | AAG: 0 | GAG: 0 | CAG: 0 | UAG: 1 | CAA: 1 | UAA: 1 |
| Phe | AAA: 0 | GAA: 1 | | | | | Phe | AAA: 0 | GAA: 1 | | | | |
| Asn | AUU: 0 | GUU: 1 | | | | | Asn | AUU: 0 | GUU: 1 | | | | |
| Lys | CUU: 0 | UUU: 1 | | | | | Lys | CUU: 0 | UUU: 1 | | | | |

Zhang et al. (2020), *PeerJ*, DOI 10.7717/peerj.10312

**Table 3** (*continued*)

| tRNA Isotypes | Isoacceptors | | | | | | tRNA Isotypes | Isoacceptors | | | | | |
|---|---|---|---|---|---|---|---|---|---|---|---|---|---|
| Asp | AUC: 0 | GUC: 1 | | | | | Asp | AUC: 0 | GUC: 1 | | | | |
| Glu | CUC: 0 | UUC: 2 | | | | | Glu | CUC: 0 | UUC: 2 | | | | |
| His | AUG: 0 | GUG: 1 | | | | | His | AUG: 0 | GUG: 1 | | | | |
| Gln | CUG: 0 | UUG: 1 | | | | | Gln | CUG: 0 | UUG: 1 | | | | |
| Ile | AAU: 0 | GAU: 0 | CAU: 1 | UAU: 0 | | | Ile | AAU: 0 | GAU: 0 | CAU: 1 | UAU: 0 | | |
| Met | CAU: 2 | | | | | | Met | CAU: 2 | | | | | |
| Tyr | AUA: 0 | GUA: 1 | | | | | Tyr | AUA: 0 | GUA: 1 | | | | |
| Cys | ACA: 0 | GCA: 1 | | | | | Cys | ACA: 0 | GCA: 1 | | | | |
| Trp | CCA: 1 | | | | | | Trp | CCA: 1 | | | | | |
| Supressor | CUA: 0 | UUA: 0 | UCA: 0 | | | | Supressor | CUA: 0 | UUA: 0 | UCA: 0 | | | |
| Sec | UCA: 0 | | | | | | Sec | UCA: 0 | | | | | |
| *R. piresii* (29) | | | | | | | *E. equisetina* (25) | | | | | | |
| Ala | AGC: 0 | GGC: 0 | CGC: 0 | UGC: 0 | | | Ala | AGC: 0 | GGC: 0 | CGC: 0 | UGC: 1 | | |
| Gly | ACC: 0 | GCC: 1 | CCC: 0 | UCC: 1 | | | Gly | ACC: 0 | GCC: 1 | CCC: 0 | UCC: 0 | | |
| Pro | AGG: 0 | GGG: 1 | CGG: 0 | UGG: 1 | | | Pro | AGG: 0 | GGG: 0 | CGG: 0 | UGG: 1 | | |
| Thr | AGU: 0 | GGU: 1 | CGU: 0 | UGU: 1 | | | Thr | AGU: 0 | GGU: 1 | CGU: 0 | UGU: 0 | | |
| Val | AAC: 0 | GAC: 1 | CAC: 0 | UAC: 1 | | | Val | AAC: 0 | GAC: 1 | CAC: 0 | UAC: 0 | | |
| Ser | AGA: 0 | GGA: 1 | CGA: 0 | UGA: 1 | ACU: 0 | GCU: 1 | Ser | AGA: 0 | GGA: 1 | CGA: 0 | UGA: 1 | ACU: 0 | GCU: 1 |
| Arg | ACG: 1 | GCG: 0 | CCG:1 | UCG: 0 | CCU: 0 | UCU: 1 | Arg | ACG: 1 | GCG: 0 | CCG:0 | UCG: 0 | CCU: 0 | UCU: 1 |
| Leu | AAG: 0 | GAG: 0 | CAG: 0 | UAG: 1 | CAA: 1 | UAA: 1 | Leu | AAG: 0 | GAG: 0 | CAG: 0 | UAG: 1 | CAA: 1 | UAA: 1 |
| Phe | AAA: 0 | GAA: 1 | | | | | Phe | AAA: 0 | GAA: 1 | | | | |
| Asn | AUU: 0 | GUU: 1 | | | | | Asn | AUU: 0 | GUU: 1 | | | | |
| Lys | CUU: 0 | UUU: 1 | | | | | Lys | CUU: 0 | UUU: 1 | | | | |
| Asp | AUC: 0 | GUC: 2 | | | | | Asp | AUC: 0 | GUC: 1 | | | | |
| Glu | CUC: 0 | UUC: 2 | | | | | Glu | CUC: 0 | UUC: 2 | | | | |
| His | AUG: 0 | GUG: 1 | | | | | His | AUG: 0 | GUG: 1 | | | | |
| Gln | CUG: 0 | UUG: 1 | | | | | Gln | CUG: 0 | UUG: 1 | | | | |
| Ile | AAU: 0 | GAU: 0 | CAU: 1 | UAU: 0 | | | Ile | AAU: 0 | GAU: 0 | CAU: 1 | UAU: 0 | | |
| Met | CAU: 2 | | | | | | Met | CAU: 2 | | | | | |
| Tyr | AUA: 0 | GUA: 1 | | | | | Tyr | AUA: 0 | GUA: 1 | | | | |
| Cys | ACA: 0 | GCA: 1 | | | | | Cys | ACA: 0 | GCA: 1 | | | | |
| Trp | CCA: 1 | | | | | | Trp | CCA: 2 | | | | | |
| Supressor | CUA: 0 | UUA: 0 | UCA: 0 | | | | Supressor | CUA: 0 | UUA: 0 | UCA: 0 | | | |
| Sec | UCA: 0 | | | | | | Sec | UCA: 0 | | | | | |

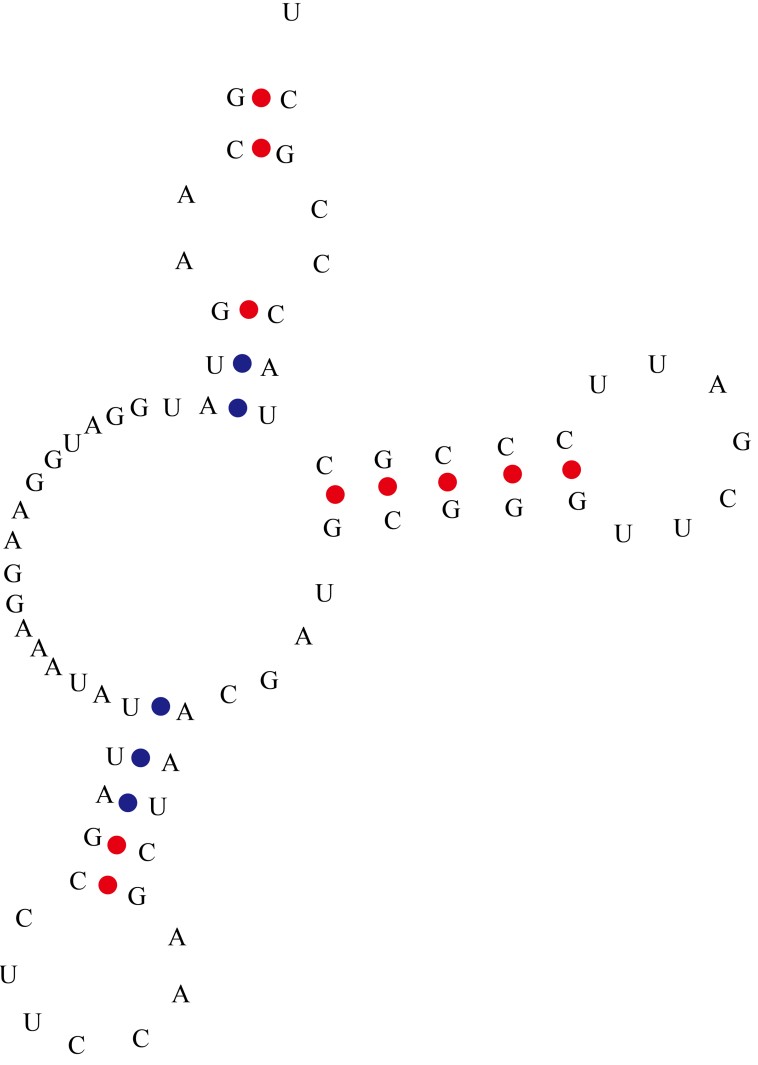

**Figure 3 An abnormal tRNA structure lacking the D-arm found in *W. nobilis*.** The tRNA*Gly* with anti-codon UCC was found lacking the D-arm.

tRNA*Gln*. In addition, in the final two positions of the Ψ-arm, all of the tRNAs were found to have conserved "G-G" nucleotides, except for tRNA*Arg*, tRNA*Cys*, tRNA*Phe*, and tRNA*Val* (Table 4). Small conserved consensus sequences were found in the Ψ region. To be specific, except for tRNA*Ser*, the Ψ-loop in tRNAs was found to contain a conserved sequence comprising U-U-C-N-A-$N_2$ according to a multiple sequence alignment of 20 members of the tRNA gene family (Table 4).

**Table 4  Conserved sequence motifs in chloroplast tRNAs from gymnosperms.** Small conserved consensus motifs were observed in the Ψ region. To be specific, except for tRNA[Ser], the Ψ-loop in tRNAs was found to contain a conserved sequence comprising U-U-C-NA-$N_2$ according to a multiple sequence alignment of 20 members of the tRNA gene family.

| tRNA Isotypes | 5′ AC-arm | D-arm | D-loop | ANC-arm | ANC-loop | Variable region | Ψ-arm | Ψ-loop |
|---|---|---|---|---|---|---|---|---|
| Alanine | 5′-G-G-G-G-A-U-A<br>IIIIIII<br>3′-C-C-C-C-U-A-U | 5′-G-C-U-C<br>IIII<br>3′-C-G-A-G | 5′-A-G-U-U-G-G-U-A | 5′-C-C-G-C-U<br>IIIII<br>3′-G-G-C-G-A | 5′-C-U-U-G-C-A-U | 5′-A-U-G-U-C | 5′-A-G-C-G-G<br>IIIII<br>3′-U-C-G-C-C | 5′-U-U-C-G-A-G-U |
| Arginine | $N_2$-G-$N_3$-G<br>IIII<br>$N_2$-C-$N_3$-C | G-$N_3$<br>II<br>C-$N_3$ | A-$N_2$-G-G-A-$N_{1-2}$-A | $N_3$-G-N<br>III<br>$N_3$-C-N | N-U-N-C-N-A-A | ***** | $N_3$-G-N<br>III<br>$N_3$-C-N | U-U-C-N-A-A-U |
| Asparagine | U-C-C-N-C-A-N<br>IIIIIII<br>A-G-G-N-G-U-N | G-C-U-C<br>IIII<br>C-G-A-G | A-G-N-G-G-U-A | G-U-C-G-G<br>IIIII<br>C-A-C-G-C | C-U-G-U-U-A-A | U-$N_{1-3}$-G-U-C | G-U-A-G-G<br>IIIII<br>C-A-U-C-C | U-U-C-A-A-A-U |
| Aspartate | G-G-G-A-U-U-G<br>IIIIIII<br>C-C-C-U-A-A-C | G-U-U-C<br>III<br>C-A-A-G | A-A-U-N-G-U-U-A | C-C-G-N-C<br>IIIII<br>G-G-C-N-G | C-U-G-U-C-A-A | A-A-G-U-U | G-C-G-G-G<br>IIIII<br>C-G-C-C-C | U-U-C-G-A-G-N |
| Cysteine | G-G-C-G-N-C-A<br>IIIIIII<br>C-C-G-C-N-G-U | G-C-C<br>III<br>C-G-G | A-A-G-N-G-G-U-A-A | G-$N_{2,3}$-A<br>III<br>C-$N_{2,3}$-U | C-U-A-C-A-A-A | **** | C-C-C-N-G<br>IIII<br>G-G-G-N-C | U-U-C-G-A-A-U |
| Glutamate | U-G-G-G-G-C-G<br>IIIIIII<br>A-C-C-C-C-G-C | G-C-C<br>III<br>C-G-G | A-A-G-N-G-G-U-A-A | N-C-A-G-G<br>IIIII<br>N-G-U-C-C | U-U-U-U-G-N-U | U-A-N-U-$N_{1-2}$ | N-A-A-G-G<br>IIII<br>N-U-U-C-C | U-U-C-G-A-A-U |
| Glutamine | G-$N_2$-C-$N_3$<br>IIII<br>C-$N_2$-G-$N_3$ | G-$N_3$<br>II<br>C-$N_3$ | A-G-N-G-G-$N_{2-4}$ | ***** | C-U-U-U-C-A-N | **** | $N_3$-G-G<br>III<br>$N_3$-C-C | U-U-C-N-A-N-U |
| Glycine | G-C-G-G-N-U-A<br>IIIIIII<br>C-G-C-C-N-A-U | G-U-U<br>III<br>C-A-A | N-A-$N_{5-14}$ | U-C-U-C-N<br>IIIII<br>A-G-A-G-N | U-U-G-C-C-A-N | A-G-A-N | G-C-G-G-G<br>IIII<br>C-G-C-C-C | U-U-C-G-A-U-N |
| Histidine | G-C-G-G-A-C-G<br>IIIIIII<br>C-G-C-C-U-G-C | G-C-C<br>III<br>C-G-G | A-A-G-U-G-G-$N_{2,4}$-A-A | G-U-G-G-A<br>IIIII<br>C-A-C-C-U | U-U-G-U-G-G-A | C-A-C-G-C | G-C-G-G-G<br>IIII<br>C-G-C-C-C | U-U-C-A-A-U-C |
| Isoleucine | G-C-A-U-C-C-A<br>IIIIIII<br>C-G-U-A-G-G-U | G-C-U-C<br>IIII<br>C-G-A-G | G-A-A-U-G-G-U-A-A-A | C-C-C-A-A<br>IIIII<br>G-G-G-U-U | C-U-C-A-U-A-A | A-A-N-U-C | G-C-N-G-G<br>IIII<br>C-G-N-C-C | U-U-C-A-A-U-U |
| Leucine | G-$N_5$-A<br>III<br>C-$N_5$-U | G-N-G<br>III<br>C-N-C | N-A-A-U-N-G-U-A-G-A | ***** | C-U-N-A-$N_2$-A | U-G-N-U-N$_{3,9}$-G-C-N-U | $N_3$-G-G<br>III<br>$N_3$-C-C | U-U-C-N-A-N-U |
| Lysine | G-G-G-U-U-G-C<br>IIIIIII<br>C-C-C-A-A-C-G | A-C-U-C<br>IIII<br>U-G-A-G | A-A-U-G-G-U-A | U-C-G-G<br>IIII<br>A-G-C-C | C-U-U-U-U-A-A | $N_2$-A-G-N-U | C-C-G-G-G<br>IIIII<br>G-G-C-C-C | U-U-C-N-A-N-U |
| Methionine | N-C-$N_{3,4}$<br>III<br>N-G-$N_{3,4}$ | **** | N$_{4,5}$-G-G-U-$N_{1,3}$ | ***** | N-U-C-A-U-A-N | $N_3$-U-C | $N_3$-G-G<br>III<br>$N_3$-C-C | U-U-C-A-A-N-U |
| Phenylalanine | G-C-C-G-G-G-A<br>IIIIIII<br>C-G-G-C-C-C-U | G-C-U-C<br>IIII<br>C-G-A-G | A-G-U-U-G-G-U-A | G-A-A-G-A<br>IIIII<br>C-U-C-C-U | C-U-G-A-A-A-A | G-U-G-C-C | A-C-C-A-G<br>IIIII<br>U-G-G-U-C | U-U-C-A-A-A-U |
| Proline | N-G-G-$N_4$<br>IIII<br>N-C-C-$N_4$ | N-C-G-C<br>IIII<br>N-G-C-G | A-G-N-U-U-G-G-U-A | ***** | N-U-N-G-G-G-U | A-N-G-U-C | $N_3$-G-G<br>III<br>$N_3$-C-C | U-U-C-A-A-N-U |
| Serine | G-$N_5$-A<br>III<br>C-$N_5$-U | G-$N_{2,3}$<br>II<br>C-$N_{2,3}$ | ********** | $N_2$-G-$N_{1,2}$<br>III<br>$N_2$-C-$N_{1,2}$ | N-U-$N_3$-A-N | **************** | G-N-G-G-G<br>IIIII<br>C-N-C-C-C | U-U-C-G-$N_2$-U |
| Threonine | G-C-$N_2$-G-$N_2$<br>IIIII<br>C-G-$N_2$-C-$N_2$ | N-C-U-C<br>IIII<br>N-G-A-G | A-G-N-G-G-U-$N_{0,1}$-A | N-C-G-C-N<br>IIIII<br>N-G-C-G-N | N-U-$N_3$-A-A | $N_2$-G-U-C | N-U-N-G-G<br>IIIII<br>N-A-N-C-C | U-U-C-N-A-$N_2$ |
| Tryptophan | G-C-G-C-U-C-U<br>IIIIIII<br>C-G-C-G-A-G-A | G-U-U-N<br>IIII<br>C-A-A-N | A-G-$N_3$-G-G-U-A | $N_2$-G-G-U<br>IIII<br>$N_2$-C-C-A | C-U-C-C-A-A-A | A-U-G-N-C | G-U-A-G-G<br>IIIII<br>C-A-U-C-C | U-U-C-A-A-A-U |
| Tyrosine | N-G-G-N-C-N-A<br>IIIIIII<br>N-C-C-N-G-N-U | N-C-$N_2$<br>III<br>N-G-$N_2$ | A-G-$N_6$-A | ***** | C-U-G-U-A-A-A | **** | $N_3$-G-G<br>III<br>$N_3$-C-C | U-U-C-N-A-$N_2$ |
| Valine | A-G-G-G-N-U-A<br>IIIIIII<br>U-C-C-C-N-A-U | N-C-U-C<br>IIII<br>N-G-A-G | A-G-$N_{4,6}$-A | U-C-N-C-N<br>IIIII<br>A-G-N-G-N | U-U-N-A-C-$N_2$ | $N_{2,3}$-G-U-C | $N_2$-C-N-G<br>IIII<br>$N_2$-G-N-C | U-U-C-N-A-$N_2$ |

**Notes.**

Note that the consensus sequences are shown from 5′ to 3′. The asterisk mark (⋆) show the absence of conserved nucleotide consensus sequence in respective region of chloroplast tRNAs. 5′ AC-arm, 5′ Acceptor arm; ANC-arm, Anti-codon arm; ANC-loop, Anti-codon loop; Ψ-arm, Pseudouridine arm; Ψ-loop, Pseudouridine loop. The short lines under the bases in anticodon loop of tRNA[Ile] are to indicate its possible modification.

## Diversification of tRNAs structures

The diverse arms and loops of tRNAs allow the regulation and control of protein translation. Each arm and loop has a specific nucleotide composition. Our analysis based on 373 tRNAs showed that the acceptor arm of chloroplast tRNAs contains 6 bp to 7 bp (Table S1). The D-arms were found to contain 3 or 4 bp generally, with a stable "G" in the initial position and "C" in the last position of the D-stem 5′ strand in most tRNAs (such as tRNA tRNA[Ala], tRNA[Asn], tRNA[Asp], tRNA[Cys], tRNA[Glu], tRNA[His], tRNA[Ile], and tRNA[Phe]). Most D-loops usually contain 7 to 11 nt with conserved "A" nucleotides at the two end locations. The anticodon arms of chloroplast tRNAs mainly contain 5 bp (90.4%). We found that 367

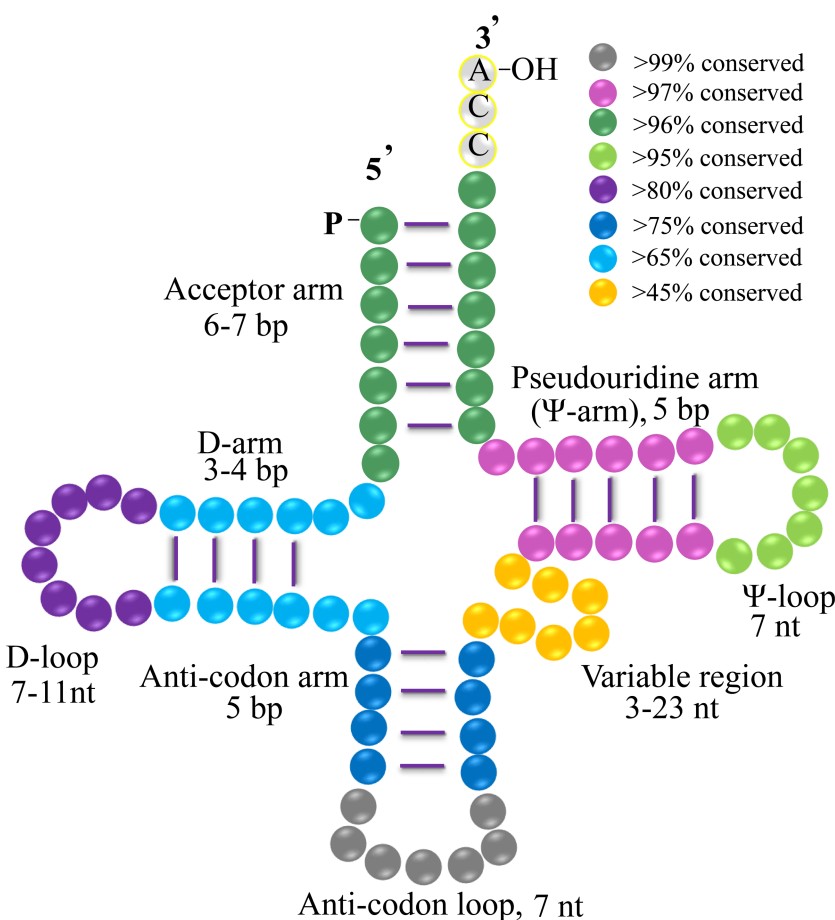

**Figure 4  Clover leaf-like structure of gymnosperms tRNA.** The tRNA contains the Acceptor arm (6–7 bp, dark green, >96% conserved), D-arm (3–4 bp, light blue, >65% conserved), D-loop (7–11 nt, purple, >80% conserved), Anti-codon arm (5 bp, dark blue, >75% conserved), anti-codon loop (7 nt, gray, >99% conserved), variable region (3–23 nt, orange, >45% conserved), Ψ-arm (5 bp, light purple, >97% conserved), and Ψ-loop (7 nt, green, >95% conserved). "% conservation" means the conservative ratio of base identities in each stem and loop structure of the whole set of gymnosperm tRNAs. Several tRNAs harbor the nucleotides of C-C-A tail.

(about 99%) tRNAs contain 7 nt in their anticodon loop, thereby indicating that the sequence of the anticodon loop is highly conserved (Table 4, Table S1). The variable loops of different tRNAs contain 3 to 23nt, where those in tRNA$^{Ala}$, tRNA$^{Asp}$, tRNA$^{His}$, tRNA$^{Phe}$, and tRNA$^{Pro}$ contain 5 bp (Table S1). The Ψ-arm contains 5 bp in most of the gymnosperm chloroplast tRNAs, except for tRNA$^{Ala}$ and some of the tRNA$^{Trp}$, tRNA$^{Gly}$, tRNA$^{Thr}$, and tRNA$^{Arg}$ in chloroplast. The Ψ-loops of most tRNAs contain 7 nt, apart from tRNA$^{Ala}$ and several of tRNA$^{Cys}$ and tRNA$^{Thr}$ (Table S1).

## Gymnosperm chloroplast tRNAs derived from multiple common ancestors

The phylogenetic tree demonstrated the presence of three major clusters covering 64 groups and the different types of all tRNAs (as shown by the different strings in Fig. S1).

We detected 37 groups in cluster I, five in cluster II, and 22 groups in cluster III. Cluster I contains tRNA tRNA$^{Ser}$, tRNA$^{Tyr}$, tRNA$^{His}$, tRNA$^{Gln}$, tRNA$^{Thr}$, tRNA$^{Pro}$, tRNA$^{Gly}$, tRNA$^{Met}$, tRNA$^{Asp}$, tRNA$^{Arg}$, tRNA$^{Ala}$, tRNA$^{Cys}$, tRNA$^{Lys}$, tRNA$^{Glu}$, tRNA$^{Ile}$, tRNA$^{Asn}$, tRNA$^{Val}$, tRNA$^{Leu}$, and tRNA$^{Trp}$. Cluster II contains tRNA$^{His}$, tRNA$^{Ser}$, tRNA$^{Tyr}$, and tRNA$^{Leu}$. Cluster III contains tRNA$^{Leu}$, tRNA$^{Ile}$, tRNA$^{Gly}$, tRNA$^{Thr}$, tRNA$^{Ser}$, tRNA$^{Val}$, tRNA$^{Glu}$, tRNA$^{Lys}$, tRNA$^{Cys}$, tRNA$^{Gln}$, tRNA$^{His}$, tRNA$^{Arg}$, tRNA$^{Phe}$, tRNA$^{Ala}$, and tRNA$^{Met}$ (Fig. S1). tRNA$^{Ser}$, tRNA$^{His}$, and tRNA$^{Leu}$ are present in cluster I but also in cluster II and cluster III, thereby suggesting that these tRNAs evolved from multiple lineages. Most of the tRNAs were found to form more than one group in the phylogenetic tree. In cluster I, the tRNAs that formed two groups in the phylogenetic tree were identified as tRNA$^{Tyr}$, tRNA$^{Gln}$, tRNA$^{Met}$, tRNA$^{Asp}$, tRNA$^{Ala}$, tRNA$^{Lys}$, tRNA$^{Ile}$, and tRNA$^{Trp}$, whereas those that clustered to form three groups were determined as tRNA$^{Ser}$, tRNA$^{Pro}$, tRNA$^{Arg}$, tRNA$^{Glu}$, tRNA$^{Asn}$, tRNA$^{Val}$, and tRNA$^{Leu}$. Moreover, tRNA$^{Thr}$ clustered into four groups. In cluster II, tRNA$^{Ser}$ was found to form two groups. In cluster III, tRNA$^{Gly}$ and tRNA$^{Val}$ were found to form two groups, whereas tRNA$^{Thr}$ formed three groups, tRNA$^{Ile}$ formed four groups. Some tRNAs in cluster III were found to group individually, where these tRNAs containing the anticodons C-G-A in tRNA tRNA$^{Ser}$, U-U-C in tRNA$^{Glu}$, U-U-U in tRNA$^{Lys}$, G-C-A in tRNA$^{Cys}$, U-U-G in tRNA$^{Gln}$, G-U-G in tRNA$^{His}$, U-C-U in tRNA$^{Arg}$, G-A-A in tRNA$^{Phe}$, U-G-C in tRNA$^{Ala}$, and C-A-U in tRNA$^{Met}$ all grouped separately (Fig. S1). The multiple groupings of different tRNAs suggest that they evolved from multiple common ancestors. Furthermore, the tRNAs presented in cluster III, i.e., tRNA$^{Met}$ (CAU), tRNA$^{Thr}$ (UGU, GGU), tRNA$^{Val}$ (UAC), tRNA$^{Ala}$ (UGC), tRNA$^{Phe}$ (GAA), tRNA$^{Arg}$ (UCU), tRNA$^{His}$ (GUG), tRNA$^{Gln}$ (UUG), tRNA$^{Cys}$ (GCA), tRNA$^{Lys}$ (UUU), tRNA$^{Glu}$ (UUC), tRNA$^{Ile}$ (UAU), tRNA$^{Val}$ (GAC), tRNA$^{Leu}$ (CAA), tRNA$^{Gly}$ (UCC), tRNA$^{Ser}$ (CGA), tRNA$^{Gly}$ (GCC), and tRNA$^{Ile}$ (CAU), tended to be the most basic tRNAs and they had undergone gene duplication and diversification to generate other tRNA molecules.

## C-A-U anticodon in tRNA$^{Ile}$

Our detailed genomic study showed that tRNA$^{Ile}$ also encodes a C-A-U anticodon in addition to the presence of this typical anticodon in tRNA$^{Met}$. In general, the C-A-U anticodon is recognized as a typical characteristic of tRNA$^{Met}$ and there is only one iso-acceptor. In particular, we found that the tRNA$^{Ile}$ in *T. mairei* encodes two C-A-U anticodons, and *C. debaoensis*, *S. verticillata*, *D. spinulosum*, *C. lanceolata*, *G. biloba*, *C. deodara*, *W. mirabilis*, *G. gnemon*, *R. piresii*, *E. equisetina*, and *W. nobilis* also encode a C-A-U anticodon (Table 3, Data S1, Fig. S3).

## Transition/transversion of tRNAs

A previous study (*Mohanta et al., 2019*) showed that the evolutionary rates are almost equal for tRNAs with respect to transition and transversion despite the low probability of transition or transversion events in tRNAs. In this study, we identified several intriguing substitutions of gymnosperm chloroplast tRNAs. Overall, our analysis of the substitution rates detected using the whole set of chloroplast tRNAs showed that average transition rate (15.38) was significantly larger than the average transversion rate (4.81) with a ratio of 3:1
**Table 5  Transition and transversion rate of chloroplast tRNA.** In all of the chloroplast tRNAs, the average transition rate (shown in bold value) was slightly higher than the average transversion rate, thereby indicating that chloroplast tRNAs have unequal substitution rates.

| From/To | A | U | C | G | From/To | A | U | C | G |
|---|---|---|---|---|---|---|---|---|---|
| | | Alanine | | | | | lysine | | |
| A | – | 12.50 | 12.50 | **0.00** | A | – | 1.47 | 1.47 | **22.06** |
| U | 12.50 | – | **0.00** | 12.50 | U | 1.47 | – | **22.06** | 1.47 |
| C | 12.50 | **0.00** | – | 12.50 | C | 1.47 | **22.06** | – | 1.47 |
| G | **0.00** | 12.50 | 12.50 | – | G | **22.06** | 1.47 | 1.47 | – |
| | | Arginine | | | | | Methionine | | |
| A | – | 2.93 | 2.93 | **19.13** | A | – | 3.86 | 3.86 | **17.27** |
| U | 2.93 | – | **19.13** | 2.93 | U | 3.86 | – | **17.27** | 3.86 |
| C | 2.93 | **19.13** | – | 2.93 | C | 3.86 | **17.27** | – | 3.86 |
| G | **19.13** | 2.93 | 2.93 | – | G | **17.27** | 3.86 | 3.86 | – |
| | | Asparagine | | | | | Phenylalanine | | |
| A | – | 1.12 | 1.12 | **22.75** | A | – | 1.21 | 1.21 | **22.58** |
| U | 1.12 | – | **22.75** | 1.12 | U | 1.21 | – | **22.58** | 1.21 |
| C | 1.12 | **22.75** | – | 1.12 | C | 1.21 | **22.58** | – | 1.21 |
| G | **22.75** | 1.12 | 1.12 | – | G | **22.58** | 1.21 | 1.21 | – |
| | | Aspartate | | | | | Proline | | |
| A | – | 0.00 | 0.00 | **25.00** | A | – | 1.53 | 1.53 | **21.95** |
| U | 0.00 | – | **25.00** | 0.00 | U | 1.53 | – | **21.95** | 1.53 |
| C | 0.00 | **25.00** | – | 0.00 | C | 1.53 | **21.95** | – | 1.53 |
| G | **25.00** | 0.00 | 0.00 | – | G | **21.95** | 1.53 | 1.53 | – |
| | | Cysteine | | | | | Serine | | |
| A | – | 2.75 | 2.75 | **19.50** | A | – | 5.16 | 5.16 | **14.68** |
| U | 2.75 | – | **19.50** | 2.75 | U | 5.16 | – | **14.68** | 5.16 |
| C | 2.75 | **19.50** | – | 2.75 | C | 5.16 | **14.68** | – | 5.16 |
| G | **19.50** | 2.75 | 2.75 | – | G | **14.68** | 5.16 | 5.16 | – |
| | | Glutamine | | | | | Threonine | | |
| A | – | 3.83 | 3.83 | **17.35** | A | – | 3.91 | 3.91 | **17.18** |
| U | 3.83 | – | **17.35** | 3.83 | U | 3.91 | – | **17.18** | 3.91 |
| C | 3.83 | **17.35** | – | 3.83 | C | 3.91 | **17.18** | – | 3.91 |
| G | **17.35** | 3.83 | 3.83 | – | G | **17.18** | 3.91 | 3.91 | – |
| | | Glutamate | | | | | Ttyptophan | | |
| A | – | 4.59 | 4.59 | **15.81** | A | – | 2.02 | 2.02 | **20.96** |
| U | 4.59 | – | **15.81** | 4.59 | U | 2.02 | – | **20.96** | 2.02 |
| C | 4.59 | **15.81** | – | 4.59 | C | 2.02 | **20.96** | – | 2.02 |
| G | **15.81** | 4.59 | 4.59 | – | G | **20.96** | 2.02 | 2.02 | – |
| | | Glycine | | | | | Tyrosine | | |
| A | – | 1.94 | 1.94 | **21.13** | A | – | 5.34 | 5.34 | **14.33** |
| U | 1.94 | – | **21.13** | 1.94 | U | 5.34 | – | **14.33** | 5.34 |
| C | 1.94 | **21.13** | – | 1.94 | C | 5.34 | **14.33** | – | 5.34 |
| G | **21.13** | 1.94 | 1.94 | – | G | **14.33** | 5.34 | 5.34 | – |
**Table 5** (*continued*)

| From/To | A | U | C | G | From/To | A | U | C | G |
|---|---|---|---|---|---|---|---|---|---|
| | | Histidine | | | | | Valine | | |
| A | – | 1.22 | 1.22 | **22.56** | A | – | 1.73 | 1.73 | **21.54** |
| U | 1.22 | – | **22.56** | 1.22 | U | 1.73 | – | **21.54** | 1.73 |
| C | 1.22 | **22.56** | – | 1.22 | C | 1.73 | **21.54** | – | 1.73 |
| G | **22.56** | 1.22 | 1.22 | – | G | **21.54** | 1.73 | 1.73 | – |
| | | Isoleucine | | | | | Overrall | | |
| A | – | 4.72 | 4.72 | **15.56** | A | – | 4.81 | 4.81 | **15.38** |
| U | 4.72 | – | **15.56** | 4.72 | U | 4.81 | – | **15.38** | 4.81 |
| C | 4.72 | **15.56** | – | 4.72 | C | 4.81 | **15.38** | – | 4.81 |
| G | **15.56** | 4.72 | 4.72 | – | G | **15.38** | 4.81 | 4.81 | – |
| | | Leucine | | | | | | | |
| From/To | A | U | C | G | | | | | |
| A | – | 4.40 | 4.40 | **16.21** | | | | | |
| U | 4.40 | – | **16.21** | 4.40 | | | | | |
| C | 4.40 | **16.21** | – | 4.40 | | | | | |
| G | **16.21** | 4.40 | 4.40 | | | | | | |

(Table 5). The same transition: transversion ratio bias was found in all the set of tRNAs for tRNA[Ser], tRNA[Glu], tRNA[Tyr], tRNA[Ile], tRNA[Met], tRNA[Gln], tRNA[Thr], and tRNA[Leu]. The ratio was over 6:1 for tRNA[Cys] and tRNA[Arg]. The transition rates for tRNA[Trp], tRNA[Val], and tRNA[Gly] were about 10 times higher than their transversion rates. These findings suggest that tRNA[Ser], tRNA[Glu], tRNA[Tyr], tRNA[Ile], tRNA[Met], tRNA[Gln], tRNA[Thr], tRNA[Leu], tRNA[Cys], tRNA[Arg], tRNA[Trp], tRNA[Val] and tRNA[Gly] underwent transition substitutions more readily than transversion substitutions during their evolution in gymnosperm chloroplast genomes. In addition, the transition rates in tRNA[Lys] and tRNA[Pro] were about 15 times higher than their transversion rates. The transition rates in tRNA[Asn], tRNA[Phe], and tRNA[His] were about 20 times higher than their transversion rates. These results indicate that tRNAs are much more likely to have undergone transition events rather than transversion events. The highest transversion rate of 12.50 was found in tRNA[Ala] and the lowest transversion rate of 0.00 in tRNA[Asp] (Table 5). Correspondingly, tRNA[Ala] lacks any transitions (Table 5).

## tRNA duplication/loss events

In addition to transition and transversion events, gene duplication and loss events have played important roles in gene evolution. Our analysis of duplication and loss events indicated that 153 duplication events (duplication and conditional duplication) have occurred in all of the gymnosperm chloroplast tRNA genes investigated in this study (Fig. S2). In addition, 220 gymnosperm chloroplast tRNA gene loss events were detected (Table S2, Fig. S2). Thus, the loss of genes was slightly more frequent than their duplication for gymnosperm chloroplast tRNA genes.

## DISCUSSION

tRNAs are major genetic components of semi-autonomous chloroplasts and our analysis of gymnosperm chloroplast genomes showed that they have several basic conserved genomic features. The gymnosperm chloroplast genomes investigated in the present study were found to encode 28 to 33 tRNA isotypes, thereby indicating that there is substantial variation in the quantity of tRNAs in gymnosperm chloroplast genomes. The lack of tRNA[Ala] in *R. piresii* and *T. mairei*, and the absence of tRNA[Val] in *T. mairei* were interesting. Thus, it is necessary to understand how the translation process is conducted in chloroplasts without these crucial tRNAs. According to previous studies (*Treangen & Rocha, 2011*; *Mohanta et al., 2019*), it is likely that the deficiency of these tRNAs is compensated for by the transfer of corresponding tRNAs from the nucleus or mitochondria. In addition to the absence of tRNA[Ala] and tRNA[Val], all of the gymnosperm plants were shown to not encode selenocysteine tRNA and its suppressor tRNA in their chloroplast genomes (Table 2). Selenocysteine tRNA and its suppressor tRNA were also not detected in the chloroplast of *Oryza sativa* (*Mohanta & Bae, 2017*).

In addition to the presence of C-A-U anticodon in tRNA[Met], we found that tRNA-CAU is present in tRNA[Ile] (Table 3). Similarly, the C-A-U anticodon was detected in tRNA[Ile] in *Bacillus subtilis* (Ehrenberg) Cohn and spinach (*Kashdan & Dudock, 1982*; *Köhrer et al., 2014*). The possible mechanism that governs the specificity of this amino acid may involve modification of the wobble position in the anticodon by a tRNA-modifying enzyme. Chloroplasts originate from bacteria so the tRNA modifications found in bacteria may also occur in chloroplast tRNAs. In bacteria, the tRNA-modifying enzyme TilS can convert the 5′-C residue in the CAU anticodon of specific tRNA[Ile] molecules into lysidine to decode 5′-AUA (Ile) codons instead of 5′-AUG (Met) codons (*Soma et al., 2003*). In addition, when lysidine decodes isoleucine, the tautomer form of lysidine provides compatible hydrogen bond donor–acceptor sites to allow base pairing with "A" and this may help to the recognition of the codon AUA instead of AUG (*Sonawane & Tewari, 2008*; *Sambhare et al., 2014*). The absence of tRNA[Ile]-lysidine synthetase leads to a failure to modify C34 to lysidine in tRNA[Ile] (LAU) (i.e., the synthesis of CAU-tRNA[Ile]) and this inactivates the translation of AUA codons (*Köhrer et al., 2014*).

During protein coding, a certain species or gene tends to use one or more specific synonym codons, which is referred to as codon usage bias (*Comeron & Aguadé, 1998*; *Rota-Stabelli et al., 2012*). In the present study, tRNA[Arg]-CCG was found to be present in the genomes of nine species but absent from *C. lanceolata*, *T. mairei*, and *E. equisetina*. Similarly, tRNA[Gly]-UCC was shown to be absent from the chloroplast genomes of *C. debaoensis*, *S. verticillata*, *D. spinulosum*, *C. lanceolata*, *T. mairei*, and *E. equisetina* (Table 3). These results suggest that gymnosperm chloroplast tRNA genes are characterized by codon usage bias (*Wei & Jin, 2017*; *Li et al., 2015*).

In general, the secondary structure of tRNAs is characterized as clover leaf-like, except for a few tRNAs with unusual secondary structures (*Jühling et al., 2018*). In our study, we identified clover leaf-like tRNAs with expanded variable loop regions (Figs. 1 and 2). Numerous tRNA[Leu], tRNA[Ser], and tRNA[Tyr] were found to have specific variable loop

configurations in terms of length and structure, suggesting significant structural variation among chloroplast tRNAs. It is interesting to note that there were also stem-loop structures in variable regions of certain tRNAs in cyanobacteria. This might indicate that similar structural variations exist between chloroplast tRNAs and cyanobacterial tRNAs (*Mohanta et al., 2017*). Future studies will have to determine the biological importance of these variant tRNAs. The novel tRNA structure lacking the D arm might play some other significative functions in the translation progress and additional research is necessary to elucidate its exact function and mechanisms. Most tRNAs have a clover-like structure formed by complementary base pairing between small segments (*Hubert et al., 1998*; *Florentz, 2002*). Previous studies have showed that the acceptor arm of tRNAs in chloroplasts contain 7 bp to 9 bp, the D-arm contains 3 bp to 4 bp, the D-loop has 4 nt to 12 nt, the anticodon arm has 5 bp, the anticodon loop contains 7 nt, the variable region comprises 4 nt to 23 nt, and Ψ-arm contains 5 bp, and the Ψ-loop has 7 nt (*Wilusz, 2015*; *Mohanta & Bae, 2017*; *Mohanta et al., 2019*). In the present study, we found that the acceptor arm of chloroplast tRNAs contains 6 bp to 7 bp in 373 tRNAs, where the D-arm has 3 bp or 4 bp and the D-loop usually contains 7 nt to 11 nt. The anticodon loop of gymnosperm chloroplast tRNAs generally contains 7 nt, and thus the sequence of the anticodon loop is typically conserved (Table 4, Table S1). The variable loop of different tRNAs contain 3 nt to 23 nt (Table S1). The Ψ-arm of gymnosperm chloroplast tRNAs generally contains 5 bp and the Ψ-loop has 7 nt (Table S1). Our results are consistent with previous findings (*Wilusz, 2015*; *Mohanta & Bae, 2017*) and they suggest that chloroplast RNAs are significantly conserved. The consensus sequence "U-U-C-N-A-$N_2$" was found in the Ψ region (Table 4). Previous studies also reported the existence of a similar sequence in the Ψ-loop of tRNAs in *Oryza sariva* and Cyanobacteria (*Mohanta & Bae, 2017*; *Mohanta et al., 2017*). This suggests that the consensus "U-U-C-N-A-$N_2$" motif of the Ψ region, identified here and in previous analyses, is a general consensus motif of canonical tRNAs.

Our phylogenetic analysis detected three clear clusters and many tRNA groups. Some tRNAs (tRNA[Ser], tRNA[His], and tRNA[Leu]) in cluster I and cluster II were also in cluster III, thereby indicating that these tRNAs evolved from multiple lineages by gene duplication and gene divergence. Moreover, anticodon types comprising CGA, UUC, UUU, GCA, UUG, GUG, UCU, UGC, and CAU appeared several times in the phylogenetic tree, and thus the corresponding tRNAs evolved from multiple common ancestors. The overlapping of tRNAs groups demonstrates that these tRNAs might have diverse common ancestors in the evolutionary process (*Mohanta & Bae, 2017*). Phylogenetic analysis also showed that tRNA[Met] (CAU), tRNA[Thr] (UGU, GGU), tRNA[Val] (UAC), tRNA[Ala] (UGC), tRNA[Phe] (GAA), tRNA[Arg] (UCU), tRNA[His] (GUG), tRNA[Gln] (UUG), tRNA[Cys] (GCA), tRNA[Lys] (UUU), tRNA[Glu] (UUC), tRNA[Ile] (UAU), tRNA[Val] (GAC), tRNA[Leu] (CAA), tRNA[Gly] (UCC), tRNA[Ser] (CGA), tRNA[Gly] (GCC), and tRNA[Ile] (CAU) in cluster III tended to be the most basic tRNAs, whereas tRNA[Met] tended to be the most original tRNA. Overall, the results clearly indicate that the tRNAs encoded in gymnosperm chloroplast genomes have multiple common evolutionary ancestors.

Our results also provided insights into the gene substitution rates in gymnosperm chloroplast tRNAs. Overall, the average transition rate for tRNAs was greater than the

transversion rate, where the relationship was about 3:1 (Table 5). In all of the chloroplast tRNAs, the average transition rate was slightly higher than the average transversion rate, thereby indicating that chloroplast tRNAs have unequal substitution rates.

In addition to the transition and transversion events in tRNAs, loss and duplication events have played significant roles in the evolution of tRNAs in gymnosperm chloroplast genomes (*He & Zhang, 2006*; *Magadum et al., 2013*). In general, the gene loss events tended to occur after whole genome duplication events. We found 153 duplication events and 220 loss events in gymnosperm chloroplast tRNAs, and thus loss events have occurred slightly more frequently than duplication events (Table S2).

## CONCLUSIONS

Our basic structure analysis showed that gymnosperm chloroplast genomes encode 25 to 30 anticodon-specific tRNAs. The acceptor arm of chloroplast tRNA contains 6 bp to 7 bp, the D-arm has 3 bp or 4 bp, the D-loop contains 7 nt to 11 nt mainly, and the anticodon loop usually contains 7 nt. In different tRNAs, the variable loop contains 3 nt to 23 nt. The $\Psi$-arm contains a conserved sequence comprising U-U-C-N-A-N$_2$. tRNA$^{Ala}$ was absent from *R. piresii* and *T. mairei*, and tRNA$^{Val}$ was lacking in *T. mairei*. Gymnosperm chloroplasts do not encode selenocysteine tRNA and its suppressor tRNA in their genomes. A CAU anticodon is encoded in tRNA$^{Met}$ as well as in tRNA$^{Ile}$. A novel tRNA structure lacking the D arm was identified for the chloroplast tRNA$^{Gly}$ of *W. nobilis*. Numerous tRNA$^{Leu}$, tRNA$^{Ser}$, and tRNA$^{Tyr}$ types were found to have expanded variable regions. Phylogenetic analysis showed that tRNAs might have multiple common ancestors in the evolutionary process. Different tRNAs harbored their own transition/transversion rates, i.e., it was iso-acceptor specific. And the transition rate was generally higher than the transversion rate. Furthermore, gene loss events (220) have occurred slightly more frequently than gene duplication events (153) in gymnosperm chloroplast tRNAs. Our results provide new insights into the evolution of gymnosperm chloroplast tRNAs and their diverse roles.

## ACKNOWLEDGEMENTS

We thank Mr. Heng Liu for his kindly help for the evolutionary analysis of chloroplast tRNA. We also thank College of Life Sciences, Northwest University for supporting device platform for this study.

### Funding

This work was financially supported by the National Natural Science Foundation of China (31970359), the Shaanxi Science and Technology Innovation Team (2019TD-012), the Public health specialty in the Department of traditional Chinese Medicine (Grants no. 2017-66 and 2018-43) and the Open Foundation of Key Laboratory of Resource Biology and Biotechnology in Western China (Ministry of Education) (Grants no. ZSK2017007

and ZSK2019008). The funders had no role in study design, data collection and analysis, decision to publish, or preparation of the manuscript.

## Grant Disclosures

The following grant information was disclosed by the authors:

National Natural Science Foundation of China: 31970359.

Shaanxi Science and Technology Innovation Team: 2019TD-012.

Public health specialty in the Department of traditional Chinese Medicine: 2017-66, 2018-43.

Open Foundation of Key Laboratory of Resource Biology and Biotechnology in Western China (Ministry of Education): ZSK2017007, ZSK2019008.

## Competing Interests

The authors declare there are no competing interests.

## Author Contributions

- Ting-Ting Zhang and Yi-Kun Hou performed the experiments, analyzed the data, prepared figures and/or tables, authored or reviewed drafts of the paper, and approved the final draft.
- Ting Yang and Ming Yue analyzed the data, authored or reviewed drafts of the paper, and approved the final draft.
- Shu-Ya Zhang analyzed the data, prepared figures and/or tables, and approved the final draft.
- Jianni Liu conceived and designed the experiments, authored or reviewed drafts of the paper, and approved the final draft.
- Zhonghu Li conceived and designed the experiments, analyzed the data, authored or reviewed drafts of the paper, and approved the final draft.

## Data Availability

tRNA sequences of gymnosperms chloroplast genome conducted in the study are available in the Supplementary File.

## Supplemental Information

Supplemental information for this article can be found online at http://dx.doi.org/10.7717/peerj.10312#supplemental-information.

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
