# Peer review of "Evolutionary analysis of chloroplast tRNA of Gymnosperm revealed the novel structural variation and evolutionary aspect"

_PeerJ, doi:10.7717/peerj.10312_

## Round 0.1 · original submission · Major Revisions

Please take the comments from reviewers into consideration for revision.

Reviewer 1 ·

Basic reporting

The structure of the manuscript is well organized and easy to follow, but the english language is not of sufficient quality.

Experimental design

The authors used the speceis tree to reconcile the tRNA phylogeny to identify gene loss and gain events. No figure legend was provided for the figure s1. What is the difference between the "D" and "cD" on the nodes of the phylogeny?

Validity of the findings

The authors claimed the phylogenetic tree of tRNAs showed the presence of three major clusters covering 64 groups of all tRNAs. How the three major clusters are recognized? This conclusion could not be obtained basing on the current form of the figure 3. The authors should provide more details about getting this conclusion.

The authors found that "tRNAAla has been found to be absent in R. piresii and T. mairei while tRNAVal has been found with inexistence in T. mairei". It would be interesting to investigate whether the absent of these tRNAs affected the codon usage of the protein coding genes in the chloroplast of these species.

Additional comments

This study aimed: (1) to determine nucleotide diversification of secondary structure of gymnosperm tRNAs; (2) to identify the detailed genomic aspects of chloroplast tRNAs; (3) to assess the evolutionary relationship between different chloroplast tRNAs; (4) to evaluate the duplication or loss on all involved tRNAs. This appeared to be original. There are a few points that need to be further clarified as mentioned above.

Reviewer 2 ·

Basic reporting

Basically, it is interesting to study the evolution of tRNAs encoded in the chloroplast genomes of gymnosperms, considering that they are maternally or paternally inherited, with evolution rates about five times slower than that of nuclear genomes. However, the manuscript is immature in its present form.

Major concerns:

a) The English of the manuscript needs to be improved substantially; the sense of many sentences is unclear, cryptic or misleading.

b) The resolution/quality of Figure 1 is too low. I am also not convinced to classify the tRNAs shown in Fig. 1 as “non-cloverleaf-like structures”. They adopt cloverleaf structures, but additionally have more or less rigid variable arms.

c) Table 4: it is not immediately clear which part of the tRNA these consensus sequences represent; for example, in the column “Ψ arm”, the reader is wondering if this is the sequence of the T stem 5’- or 3’-strand? Indicate this more clearly, also give sequence polarity. The authors should include a tRNA secondary structure where the mark the sequence elements specified in Table 4.

d) These endless listings of tRNAs (for example, in lines 265 to 313) makes the manuscript hard to digest. The authors should illustrate their findings in additional summary figures. The clustering of tRNAs is not comprehensible for the reader. The authors should exemplarily delineate the sequence criteria underlying the clustering of the respective tRNA; explain why some tRNAs appear in more than one group. The evolutionary clustering of gymnosperm chloroplast tRNAs, cumulating in the phylogenetic tree of Figure 3, is at the heart of this story and should thus be more intelligible to the reader.

e) The CAU anticodon of tRNA-Ile: it is known from bacteria that the enzyme TilS converts the 5’-C residue in the CAU anticodon of specific tRNA-Ile molecules to lysidine (2-lysyl cytidine; abbreviated as L or k2C ) to decode 5’-AUA (Ile) codons instead of 5’-AUG (Met) codons (Reference: Soma A, Ikeuchi Y, Kanemasa S, Kobayashi K, Ogasawara N, Ote T, Kato J, Watanabe K, Sekine Y, Suzuki T: An RNA-modifying enzyme that governs both the codon and amino acid specificities of isoleucine tRNA. Mol Cell 2003, 12(3):689–698). This mechanism may also be operational in chloroplasts. The authors should discuss this issue and should check the literature if there is evidence for the same mechanism in chloroplasts.

f) Figure 2: describe this predicted W. nobilis RNA structure more cautiously as a tRNA-Gly-like structure, as it has a (most likely) non-functional acceptor stem that is not charged with glycine.

Experimental design

Difficult to assess at present.

Validity of the findings

Difficult to assess at present.

Reviewer 3 ·

Basic reporting

See comments to author.

Experimental design

See comments to author.

Validity of the findings

See comments to author.

Additional comments

Here are the comments which can be useful but not limited in improving the structure and quality of the manuscript.

1. Authors did not talk about the presence of modified bases in tRNA?

2. The English used in the manuscript is very poor and needs to be improved heavily so as to reach an average science audience. It was very hard to understand at multiple places that the reader loses connectivity with the theme of manuscript. Use of few words (e.g. besides) at unnecessary places needs to be avoided. Please avoid long and complex sentences such as line no. 76-80. Also, multiple sentences are repeating in every section. Because of poor English, at many places there arises a doubt on the experiments performed. So the whole manuscript has to be restructured.

3. All the figures need to be submitted online with proper fig captions. Please rename fig 1b as 2 and so on if possible. Fig. 3 for the phylogenetic tree is not at all clear hence I could not validate the results based on it. The manuscript contains most of the data based on that figure. Hence, at this stage the results cannot be validated.

4. It will be useful to include a line or two in the abstract or introduction stating the importance of this study or how these results are promising over the past studies.

5. The experimental protocol needs to be explored in more precise stepwise manner (probably English improvisation could help).

6. Line no. 138 says bacterial parameter in tRNAScan-SE. Why this was used?

7. Protocol described for phylogenetic tree construction from line no. 144 is unclear and its present written form will be difficult to reproduce the results. Also from line no. 161 various methods were used to filter the data. What is the rationale behind such multiple filtering? How each filter can be correlated in case of this gymnosperm tRNA analysis?

8.Please include the results of gene manipulation (line no. 169) in the main document as they are much essential to interpret the results stated in the manuscript.

9. Line no. 265-313 & 424-438 is totally un-interpretable because of poor resolution of fig 3. Also, line no. 349-355 correspond to fig S1 and table S2. Please make them available in the main text.

10. For this line, “Notwithstanding, it is still unclear that how the tRNA captured with CAU anti-codon to distinguish Methionine and Isoleucine and carry them respectively” authors may look into these references; Nucleosides, Nucleotides and Nucleic Acids. 27, 1158-1174, 2008; RSC Advances, 4, 14176- 14188, 2014;

11. Please provide a reference to the statement made in line no. 377.

12. Line no. 397 mentions a novel tRNA structure lacking the acceptor arm. I would like to know how the authors have re-validated its occurrence. Please explain.

12. The conclusions do not completely reflect the findings. Please fortify this section by correlating the key findings from results sections.

---

## Round 0.2 · Major Revisions

Major revision is still needed to improve the manuscript.

Reviewer 1 ·

Basic reporting

I satisfied with the revision of the manuscript, and recommended it to be accepted for publication as it stand.

Experimental design

I satisfied with the revision of the manuscript, and recommended it to be accepted for publication as it stand.

Validity of the findings

I satisfied with the revision of the manuscript, and recommended it to be accepted for publication as it stand.

Additional comments

I satisfied with the revision of the manuscript, and recommended it to be accepted for publication as it stand.

Reviewer 2 ·

Basic reporting

Review on Ms. #43190, 1st revision

1. Basic reporting:
The manuscript has improved to some extent, but there are still some unclear major and minor points. As this is a detailed bioinformatic study, the authors have to convince the reader that the analyses were meticulously carried out, taking into account that the reader cannot examine the correctness of every detail. However, at several positions in the text doubts are created whetherthe study was performed with maximum diligence, as some of the described findings and conclusions are either misleading or not comprehensible. Ín my previous review, I already criticized the endless listings of tRNAs in the text. The authors should illustrate their findings in additional summary figures in the main manuscript. I have made proposals for Fig. 4 and Table 4 (see below). Very helpful would also be a sequence/structure alignment of all analyzed gymnosperm chloroplast tRNAs (e.g. as in the tRNAdb).

Major concerns:

a) At multiple positions in the text, gaps between separate words are lacking.

b) for censensus sequence motifs, use “N” instead of “X” to indicate any of the four bases, or further differentiate (R, Y, S, W, K and so forth).
c) I find the Introduction somewhat too long in the second part describing the basics of tRNA sequence and structural elements and their functions.
d) lines 199/200, “… contain a loop structure that is similar to the anti-codon loop in the variable region of tRNAs.” This is misleading. What you want to say is that there are tRNAs with expanded variable loops, some of which form extended stem-loop structures, some shorter stem-loop structures and others are void of base-pairing within the variable loop. This is also a feature of corresponding bacterial tRNA species. My proposal for rewriting in line 199: “As shown in Fig. 1 and Fig. 2, tRNALeu, tRNASer, and tRNATyr contain expanded variable (stem-)loops.” I does not make sense to say that these variable loops are similar to the anticodon loop.
Also, some of the variable loop structures in Fig. 1 and 2 are incorrect: in Fig. 1F, a loop with a single bp in the stem is unliekley to form; likewise, in Fig. 2F, a stem without apical loop does not form. In Fig. 2B, the variable loop stem can be extended to 4 bp.
e) lines 201/202: tRNASer-GCU of D. spinulosum does not follow the consensus N-U-N-G-A-A-N. In lines 202-206, the authors should better differentiate between variable loop region (the entire sequence), and stem-loop elements in the variable loop region. I propose to rewrite from line 202 on: “The variable loop region is predicted to fold into stem-loop structures with apical loops of 3 to 7 nt in tRNASer and several tRNALeu variants. The stems contain up to 7 bp (Fig. 1 and 2). The expanded variable loop structures may play important functions during the protein translation process in chloroplasts.” At some point, these structural features should be compared with those of bacterial counterparts.
f) line 224, Tables 2 and 3: for C. lanceolata, two tRNATrp isoacceptors are indicated in Table 2, but only 1 in Table 3.
g) lines 230-235 (“A” is present in the first and the last position of the D-loop): this is not true for tRNAGln, Gly, Ile, Leu, Met. /

(In addition, in the final two positions of the Ψ-arm, all of the tRNAs were found to have conserved “G-G” nucleotides, except for tRNAArg): according to Table 4, this is also not the case for tRNACys, Phe and Val.

h) lines 233-235: it is unclear here if you found the consensus U-U-C-X-A-X2 only in a multiple sequence alignment of 20 members of the tRNA gene family or in all tRNAs analyzed here; please clarify.

i) lines 238-251: generally use “bp” when describing the length of stems; line 240: an acceptor stem of 1 or 2 bp is non-existent. In Fig. 3, the acceptor stem of W. nobilis tRNAGly likely forms a labile 7-bp acc. stem with two non-canonical A-C bp. This should also be changed in Table S1, replace “2” by “7” or “(7)” in the column “AC-arm” for W. nobilis tRNAGly. How does the AC-arm for tRNAThr of Sciadopitys_verticillata_94443 (1 bp) look like?. Provide a legend to Table S1 where you define how you counted the base pairs in the AC-arm; how do the AC-arms with 3 or 6 bp look like?

j) line 242, C in the last position of the D-arm: is this true for all tRNAs?

k) Table 4: here, the sequence of both strands of the stems should be shown; I assume that the entire sequences of the loops are listed in this table. Use a presentation such as in the tRNAdb.
Change the title to “Conserved sequence motifs in chloroplast tRNAs from gymnosperms”; what did you mean by “selected gymnosperms? All those anlyzed here?

l) line 307, correct: “… the highest transversion rate of 12.5 was found in tRNAAla and the lowest transversion rate of in tRNAAsp (Table 5). Correspondingly, tRNAAla lacks any any transitions (Table 5).”

m) line 328-330: it is unclear what you mean here; are you alluding to the possibility that these missing tRNAs are encoded in the nuclear genome and are imported into the chloroplast? Might there be tRNA editing in chloroplasts?

n) lines 345-349: please rewrite, not clear what you want to express here beyond what is expressed before.

o) line 371: there is no tRNA with a 2-bp acceptor stem (see above); please rewrite.

p) lines 378/379: the consensus U-U-C-X-A-X2” motif in the Ψ region: isn’t this a general consensus motif of canonical tRNAs? Please discuss this point.

q) line 423: what do you mean by “Transition and transversion analyses of tRNAs indicated that tRNAs are iso-acceptor specific”? Do you mean that transition/transversion patterns were isoacceptor-specific?

r) Fig. 4 is too trivial: here the authors should integrate into the tRNA 2D structure the conservations they found in their gymnosperm chloroplast tRNA set, differentiating between base identities that are e.g. >50%, > 75% and 100% conserved.

s) add legends to supplementary figures and tables with explanations for the reader.


Minor comments:

- line 209, rewrite: “The genomes of the species analyzed were found to code for at least two copies of tRNAMet-CAU/tRNAfMet-CAU. Each of the gymnosperm chloroplast genomes encodes 25 to 31 anticodon-specific tRNAs (Tables 2 and 3), where E. equisetina …”
- line 224, rephrase: “Two tRNATrp isoacceptors are present in E. equisetina chloroplasts, compared with a single one in the other gymnosperm species analyzed in this study.”

- line 228: separate paragraph heading from text.

- Table 4, footnote 2, rephrase: “Note that the consensus sequences are shown from 5’ to 3’.”

- line 270, rewrite: “… tRNAs in cluster III were found to group individually, where these tRNAs …”

- line 333_ “… in the chloroplast of Oryza …”

- line 359 ff., rewrite: “In our study, we identified cloverleaf-like tRNAs with expanded variable loop regions (Fig. 1 and 2). Numerous tRNALeu, tRNASer, and tRNATyr were found to have specific variable loop configurations in terms of length and structure, suggesting significant structural variation among chloroplast tRNAs. Future studies will have to determine the biological importance of these variant tRNAs. Noteworthy, a novel tRNA structure lacking the D arm was found for tRNAGly in W. nobilis (Fig. 3).” (As said before, compare with bacterial tRNAs)

- line 378, rewrite: “The consensus sequence “U-U-C-X-A-X2” was found in the Ψ region (Table 4).”

- line 385: “… appeared several times in the phylogenetic tree, and thus …”

- line 414, rewrite: “A CAU anticodon is encoded in tRNAMet as well as in tRNAIle. A novel tRNA structure lacking the D arm was identified for the chloroplast tRNAGly of W. nobilis. Numerous tRNALeu, tRNASer, and tRNATyr types were found to have expanded variable regions, forming stem-loop structures with up to 7 bp in tRNAsSer.”

Experimental design

Requires improved illustration of results

Validity of the findings

expand comparison of gymnosperm chloroplast tRNA features with those of bacterial tRNAs in the discussion

---

## Round 0.3 · Minor Revisions

Please revise your manuscript according to the comments from reviewers.

Reviewer 2 ·

Basic reporting

See attached PDF

Experimental design

No comment

Validity of the findings

No comment

Additional comments

See attached PDF

Annotated reviews are not available for download in order to protect the identity of reviewers who chose to remain anonymous.

Reviewer 3 ·

Basic reporting

1. In figure 1 and 2; it would be good to label the expanded variable loop of tRNAs by making coloured box just to differentiate with other loop regions.
2. In fig 3; at the hinge region it looks that there is one extra ‘A’ or its counterpart base is missing. Just check?
6. In the caption of Fig S1; it should be multiple instead of multiply.
7. In table 4; There must be some modified bases in the tRNA; especially in the anticodon loop of tRNAs; so it would be good to mention ‘symbols’ of modified bases at 32, 34 and 37 as well as other positions if they are present.

Experimental design

9. Overall, the study is good and can be published in Peer J with some minor revisions.

Validity of the findings

No comment

Additional comments

Manuscript title: “Evolutionary analysis of chloroplast tRNA of Gymnosperm revealed the novel structural variation and evolutionary aspect”
Manuscript ref Number: #43190-v2
Comments to authors (Manuscript ref Number: #43190-v2)
Dear Editor,
Herewith, I am submitting review report on the manuscript entitled “Evolutionary analysis of chloroplast tRNA of Gymnosperm revealed the novel structural variation and evolutionary aspect” (Manuscript ref Number: #43190-v2). This is the revised version of this manuscript which deals with computational analysis of genomic data of chloroplast tRNA from various gymnosperms in order to assess primarily the structural variations, nucleotide variation, consensus sequence analysis and phylogenetic analysis with respect to study base substitution and tRNA gene manipulation.
1. In figure 1 and 2; it would be good to label the expanded variable loop of tRNAs by making coloured box just to differentiate with other loop regions.
2. In fig 3; at the hinge region it looks that there is one extra ‘A’ or its counterpart base is missing. Just check?
3. In the material & method section; line no. 127-128; please mention BLAST properly.
4. In these lines; 136-138; authors have mentioned default parameters of ARGAON; in which they have mentioned as; taken sequence source as a bacterial? Why?
5. Authors sat that they have observed a novel tRNAGly in W. nobilis (Fig. 3) lacking DHU loop; it would be good to speculate the probable significance of this tRNA for the general readers.
6. In the caption of Fig S1; it should be multiple instead of multiply.
7. In table 4; There must be some modified bases in the tRNA; especially in the anticodon loop of tRNAs; so it would be good to mention ‘symbols’ of modified bases at 32, 34 and 37 as well as other positions if they are present.
8. In the conclusion section few results are mentioned, so it would be good to remove these results part and keep only conclusion of this study. By doing this authors can also reduce the size of the conclusion which is unnecessarily large and mention.
9. Overall, the study is good and can be published in Peer J with some minor revisions.

---

## Round 0.4 · Minor Revisions

Please make changes according to the comments from reviewer.

Reviewer 2 ·

Basic reporting

Three minor points should be changed before publication.

a) line 316: If I understand it correctly the sentence should be changed to: "According to previous studies (Treangen & Rocha, 2011; Mohanta et al., 2019), it is likely that the deficiency of these tRNAs is compensated for by the transfer of corresponding tRNAs from the nucleus or mitochondria."

b) in Table 4, first lines, Alanine tRNA: the 3' or 5' should not be moved to a next line; it should remain attached to the sequence; if this is a space problem, the authors may also consider to indicate only 5'- or 3'-end.

c) Fig. 4: in the legend to the figure, the authors should explain more clearly what "% conservation" means. If you consider the whole set of gymnosperm tRNAs, there cannot be > 95% conservation (green) at all positions in the acceptor stem and T arm; or is this the conservation you find for the individual gymnosperm tRNA isotypes? Please clarify.

Experimental design

no comment

Validity of the findings

no comment

---

## Round 0.5 · accepted · Accept

The authors made changes according to reviewers' comments.